# GUMP: Alleviating Oversquashing with Unitary Message Passing

## Abstract

Message passing mechanism contributes to the success of GNNs in various applications, but also brings the oversquashing problem. Recent works combat oversquashing by improving the graph spectrums with rewiring techniques, disrupting the original graph connectivity, and having limited improvement on oversquashing in terms of oversquashing measure. Motivated by unitary RNN, we propose Graph Unitary Message Passing (GUMP) to alleviate oversquashing in GNNs by applying a unitary adjacency matrix for message passing. To design GUMP, a transformation is first proposed to equip general graphs with unitary adjacency matrices and keep their original graph connectivity. Then, the unitary adjacency matrix is obtained with a unitary projection algorithm, which is implemented by utilizing the intrinsic structure of the unitary adjacency matrix and allows GUMP to be permutation-equivariant. In experiments, GUMP is incorporated into various GNN architectures and the extensive results show the effectiveness of GUMP on various graph learning tasks.

## 1 Introduction

Graph neural networks (GNNs) (Scarselli et al., 2008) have been widely used in various applications, such as social network (Fan et al., 2019) and knowledge graphs (Schlichtkrull et al., 2018). The most popular GNNs follow the message passing mechanism (Gilmer et al., 2017) to update the node representations, where each node aggregates feature vectors of its neighbors to compute its new feature vector. The message-passing mechanism is designed to be permutation-equivariant, allowing GNNs to work with graphs that have varying node orders. Currently, GNNs with message passing mechanisms have demonstrated success in various graph learning tasks, such as node classification (Kipf & Welling, 2016a), graph classification (Ying et al., 2018), and link prediction (Kipf & Welling, 2016b).

However, the message passing mechanism also inevitably brings the oversquashing problem to GNNs (Alon & Yahav, 2020; Topping et al., 2022; Banerjee et al., 2022). The oversquashing problem draws inspiration from a similar phenomenon observed in RNNs when learning long-range sequences, as noted by Alon & Yahav (2020). It refers to the situation where, as larger neighborhoods are considered, information from distant interactions funneled through specific bottlenecks minimally influences GNN training. This phenomenon involves compressing information from potentially an exponentially large number of nodes (relative to the number of layers) into fixed-sized node vectors.

Various techniques are proposed to alleviate the oversquashing problem. Topping et al. (2022) propose the Jacobian of GNN to measure oversquashing, which motivates a rewiring method that increases the curvature of the edges in a graph. Most works combat oversquashing via methods depending on the graph spectrum (i.e., the eigenvalue of the adjacency matrix). In these works, rewiring techniques increase the spectral gap by flipping edges (Banerjee et al., 2022), adding edges (Karhadkar et al., 2023), re-weighting the edges (Arnaiz-Rodríguez et al., 2022), or using expander for message passing (Deac et al., 2022). Except for increasing spectral gap, recent works bound the measure in Topping et al. (2022) with effective resistance (Black et al., 2023) and commute time (Di Giovanni et al., 2023), and propose rewiring techniques to improve these bounds. Except for improving graph spectrum, Barbero et al. (2023) propose rewiring methods to greedily add edges to increase the number of walks in a graph.

The rewiring techniques above, even motivated from different perspectives, can be justified by the Jacobian measure of oversquashing in Topping et al. (2022). For instance, increasing the spectral

gap or effective resistance can be viewed as an indirect method of improving the entries of powers of the adjacency matrix in the Jacobian measure. As a result, these rewiring methods have limited or uncertain improvements on the Jacobian measure because some of them do not directly improve it (Karhadkar et al., 2023; Black et al., 2023) or improve it in a greedy way (Barbero et al., 2023). Moreover, the rewiring techniques disrupt the original graph connectivity, resulting in the loss of crucial structural inductive biases in graph learning tasks (Battaglia et al., 2018). This makes the rewiring techniques inadequate for oversquashing. Detailed related works on oversquashing are summarized in Appendix B.

In this paper, we propose a new *one-hop message passing mechanism*, called Graph Unitary Message Passing (GUMP), to alleviate oversquashing. Motivated by existing analysis on oversquashing of RNNs (Pascanu et al., 2013; Jing et al., 2017), the measures of oversquashing in RNNs and GNNs share similar forms (Fig. 1(a)), i.e., the powers of feature transformation matrix in RNN and the powers of adjacency matrix in GNN (Section 2.1). Since the unitary parameterization of the transformation matrix has proved to be effective in capturing long-range interactions (Arjovsky et al., 2016) in RNN, we consider imposing unitarity on the adjacency matrix in GNN for message passing. With a unitary adjacency matrix for message passing, the Jacobian measure of oversquashing will not change exponentially, thereby alleviating oversquashing. Compared to existing rewiring methods (Table 1), GUMP is a general message-passing mechanism that can be applied to various GNN architectures and paves a new way for alleviating oversquashing, which achieves optimal Jacobian measure and preserves the original graph connectivity.

To design GUMP, we first propose a graph transformation algorithm in Section 2.2 to equip a general graph with unitary adjacency matrices and preserve its original graph connectivity at the same time. The transformation algorithm is based on the theory showing that unitary adjacency matrices exist for the line graph of an Eulerian graph. Then, we propose an algorithm to calculate the unitary adjacency matrix in Section 2.3. The algorithm is designed to allow GUMP to be permutation-equivariant and is implemented by utilizing the intrinsic structure of unitary adjacency matrices. Then, we propose the framework that applies GUMP to different GNN architectures in Section 3. Finally, we evaluate GUMP on several graph learning tasks in Section 4. In summary, our paper has the following contributions:

- GUMP is a new *one-hop message passing mechanism* that alleviates oversquashing by applying a unitary adjacency matrix for message passing. Compared with previous works, GUMP achieves the optimal Jacobian measure of oversquashing.

- GUMP maintains the original graph connectivity with a graph transformation algorithm and preserves the permutation equivariance of message passing with unitary projection.

- Extensive results show the effectiveness of GUMP. Further analysis of the Jacobian measure also validates GUMP's ability to alleviate oversquashing.

**Notations** In this paper, we use bold uppercase letters $\mathbf{X}$ to denote matrices, bold lowercase letters $\mathbf{x}$ to denote vectors, and lowercase letters $x$ to denote scalars. Given a matrix $\mathbf{X}$, the $i$-th row of matrix $\mathbf{X}$ is denoted as $\mathbf{x}_i$ and the entry of the $i$-th row and $j$-th column of matrix $\mathbf{X}$ is denoted as $\mathbf{X}_{ij}$. The transpose and conjugate transpose of matrix $\mathbf{X}$ is denoted as $\mathbf{X}^\top$ and $\mathbf{X}^\dagger$, respectively. A graph with $n$ nodes and $e$ edges is denoted as $G = (V, E, \mathbf{X})$ where $V = \{1, 2, \cdots, n\}$ is the node set, $E \subseteq V \times V$ is the edge set, and $\mathbf{X} \in \mathbb{R}^{n \times d}$ is the $d$-dimensional node feature matrix. For convenience, the operator $\mathsf{V}[G]$ and $\mathsf{E}[G]$ are used to denote the node set and edge set of graph $G$ respectively, i.e., $\mathsf{V}[G] = V$ and $\mathsf{E}[G] = E$. The adjacency matrix of graph $G$ is denoted as $\tilde{\mathbf{A}}[G] \in \{0, 1\}^{n \times n}$ where $\tilde{\mathbf{A}}_{ij} = 1$ if $(i, j) \in E$ and $\tilde{\mathbf{A}}_{ij} = 0$ otherwise. The normalized adjacency matrix of graph $G$ is denoted as $\hat{\mathbf{A}}[G] \in \mathbb{R}^{n \times n}$ where $\hat{\mathbf{A}}[G] = \mathbf{D}^{-1/2}\tilde{\mathbf{A}}[G]\mathbf{D}^{-1/2}$ and $\mathbf{D}$ is the degree matrix of graph $G$. We also use matrix $\mathbf{A}[G] \in \mathbb{R}^{n \times n}$ to represent the general adjacency matrix in graph $G$, i.e., $\mathbf{A}_{ij} \neq 0$ if $(i, j) \in E$ and $\mathbf{A}_{ij} = 0$ otherwise. Therefore, without specifying the type of adjacency matrix, $\mathbf{A}[G]$ can also represent $\tilde{\mathbf{A}}[G]$ and $\hat{\mathbf{A}}[G]$. For convenience, the adjacency matrices above are denoted as $\tilde{\mathbf{A}}$, $\hat{\mathbf{A}}$, and $\mathbf{A}$ respectively. Some preliminaries used in this paper are provided in Appendix C. Finally, the GNN representation at layer $k$ is denoted as $\mathbf{H}^{(k)} \in \mathbb{R}^{n \times d}$ with $d$ being the dimension of node features, and the vector $\mathbf{h}_i^{(k)} \in \mathbb{R}^d$ denotes the GNN representation of node $i$ at layer $k$.

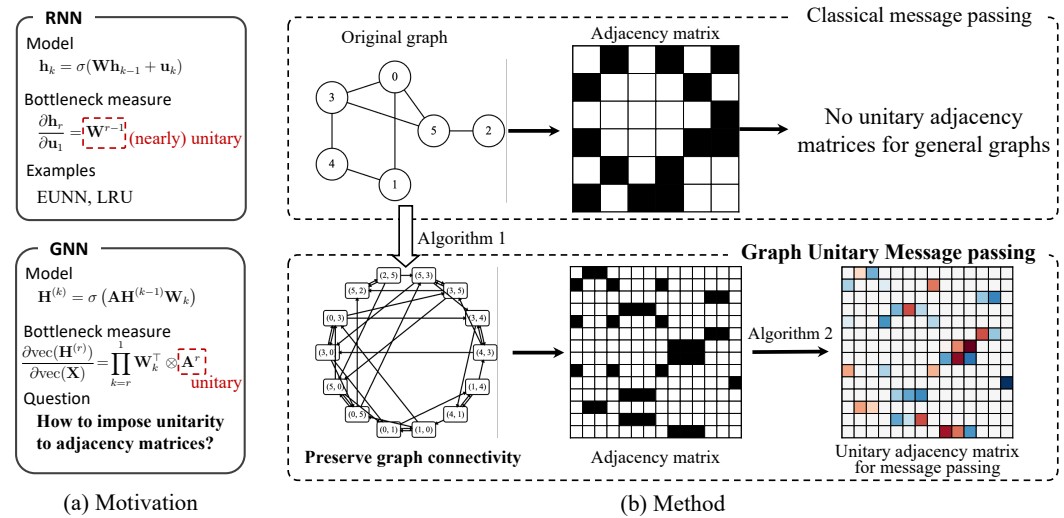

(a) Motivation            (b) Method

Figure 1: Overview of GUMP. (a) RNN versus GNN. The measures of the bottleneck are derived with the identity activation function and $\otimes$ denotes the Kronecker product. (b) GUMP aims to impose unitarity to $\mathbf{A}$ while keeping the original graph connectivity. GUMP achieves this goal with graph transformation (Section 2.2) and unitary adjacency matrix calculation (Section 2.3).

## 2 GRAPH UNITARY MESSAGE PASSING

In this section, we propose a new *one-hop message passing mechanism* called Graph Unitary Message Passing (GUMP) to alleviate oversquashing in GNNs. For simplicity, we consider the undirected graph in this paper. The extension to digraph is straightforward in Appendix F.

### 2.1 OVERVIEW

**Motivation** Unitarity has been crucial for the success of RNNs in effectively learning long-range sequences in recent years, ranging from unitary RNNs (Jing et al., 2017; Arjovsky et al., 2016) to linear RNNs (Orvieto et al., 2023; De et al., 2024). In Fig. 1(a), we simply formulate RNN as $\mathbf{h}_k = \sigma(\mathbf{W}\mathbf{h}_{k-1} + \mathbf{u}_k)$ with $\mathbf{h}_k$ being the hidden state at layer $k$, $\mathbf{W}$ being the transformation matrix, $\mathbf{u}_k$ being the input at time step $k$, and $\sigma$ being the activation function. Unitary RNN (e.g., EUNN (Jing et al., 2017)) imposes unitarity to $\mathbf{W}$ such that the gradient of long-range information does not vanish or explode, thus helps learn long-range interactions with gradient-based optimization. Recently, motivated by state space model (Gu et al., 2021), linear RNN (e.g., LRU (Orvieto et al., 2023)) uses the identity activation function and initializes $\mathbf{W}$ close to a unitary matrix to avoid the vanishing gradient problem. The success of imposing unitarity in RNNs for learning long-range sequences motivates us to apply unitarity to GNNs to alleviate oversquashing.

For graphs, it is also difficult for GNN to learn long-range interactions. In Fig. 1(a), a one-hop MPNN layer for GNN is formulated as $\mathbf{H}^{(k)} = \sigma(\mathbf{A}\mathbf{H}^{(k-1)}\mathbf{W}_k)$ for simplicity. Similar to RNN (Pascanu et al., 2013), the bottleneck of GNN is measured by the Jacobian measure $\partial\mathbf{H}^{(r)}/\partial\mathbf{x}$, which is the Jacobian of the hidden state with respect to the input (Topping et al., 2022). Therefore, we are motivated to impose unitarity to $\mathbf{A}$ in GNN to alleviate oversquashing.

**Challenges of imposing unitarity** Imposing unitarity to an adjacency matrix is not as straightforward as that in RNNs. The first challenge comes from the sparsity of the adjacency matrix, as almost all unitary matrices are dense (Fig. 1(b)). Therefore, the amount of graphs with unitary adjacency matrices is limited. The second challenge is to obtain an input-dependent unitary adjacency matrix while preserving permutation-equivariance. This is because the unitary adjacency matrix depends on the input graph, and the order of nodes in a graph should not impact GNN representations.

**GUMP** In GUMP, the first challenge is addressed by transforming the original graph to a special graph (Algorithm 1) which is guaranteed to have unitary adjacency matrices and preserves the original graph connectivity at the same time. The second challenge is addressed by calculating the unitary adjacency matrix with a unitary projection algorithm (Algorithm 2), which is implemented by utilizing the block diagonal structure of the unitary adjacency matrix and allows GUMP to be permutation-equivariant (Proposition 2.4). As a general one-hop message-passing mechanism, any convolution operation can be combined with GUMP by setting the edge weights to be the entries of the unitary adjacency matrix for message passing. The overview of GUMP is shown in Fig. 1(b).

From the theoretical perspective, GUMP has also proved to alleviate oversquashing in GNNs. Given the ReLU activation function and convolution operation in Fig. 1(a), we have the following theorem (proved in Appendix D) guaranteeing that GUMP can alleviate oversquashing.

**Theorem 2.1.** *The expected Jacobian measure for GUMP, i.e.,* $\mathbf{A}$ *is unitary in GNN, is approximately in the order of* $\mathbb{E}\big[\partial \mathbf{h}_i^{(L)}/\partial \mathbf{x}_s\big] = \mathcal{O}(1)$.

Theorem 2.1 does not indicate the Jacobian measure of GUMP is independent of $L$. In fact, the relation between the Jacobian measure and $L$ is a trigonometric function in GUMP, which can be bounded by constants. The theorem shows that the Jacobian measure of GNN with unitary adjacency matrices will not change exponentially with the number of layers, thus avoiding oversquashing. In the following sections, we will introduce the details of GUMP, including the graph transformation algorithm in Section 2.2 and the unitary adjacency matrix calculation algorithm in Section 2.3.

## 2.2 GRAPH TRANSFORMATION: CONVERT GRAPH TO HAVE UNITARY ADJACENCY MATRIX

Since unitary matrices are generally non-symmetric, the graph transformation algorithm should convert the original graph to a directed graph. We first formally define the unitary adjacency matrix as in Severini (2003). Given an adjacency matrix $\mathbf{A}$, its support matrix $\mathsf{S}[\mathbf{A}] \in \mathbb{R}^{n \times n}$ is a binary matrix with entries equal to one if the corresponding entry of $\mathbf{A}$ is non-zero and equal to zero otherwise, i.e., $\mathsf{S}[\mathbf{A}]_{ij} = 1$, if $\mathbf{A}_{ij} \neq 0$ and $\mathsf{S}[\mathbf{A}]_{ij} = 0$, if $\mathbf{A}_{ij} = 0$. Then, the unitary adjacency matrix $\mathbf{U}_G$ of graph $G$ is a unitary matrix whose support is equal to the support of its adjacency matrix $\mathbf{A}$, i.e., $\mathsf{S}[\mathbf{U}_G] = \mathsf{S}[\mathbf{A}]$.

We propose the transformation in Algorithm 1 for undirected graph $G$ to make it have unitary adjacency matrices and preserve its original graph connectivity. Algorithm 1 first transforms the undirected graph to be Eulerian graph $G'$ (Definition C.2) by splitting each undirected edge into two directed edges. Then, it converts the Eulerian graph to its line graph $\mathsf{L}(G')$ (Definition C.1). Finally, $\mathsf{L}(G')$ has unitary adjacency matrices which is proved in the following proposition.

---

**Algorithm 1** Graph transformation

**Require:** A undirected graph $G = (V, E)$;
    Initialize a new digraph $G' = (V, E')$;
    **for** $(i, j) \in E$ **do**
        Add $(i, j)$ and $(j, i)$ to $E'$;
    **end for**
    Convert $G'$ to its line graph $\mathsf{L}(G')$;
    **Return:** A digraph $\mathsf{L}(G')$.

---

**Proposition 2.2.** *The line graph* $\mathsf{L}(G')$ *returned by Algorithm 1 have unitary adjacency matrices.*

Proposition 2.2 is proved in Appendix D. In Algorithm 1, the original graph connectivity is preserved because the splitting of an undirected edge and the conversion to the line graph do not introduce new connectivity between nodes that is absent in the original graph. Finally, Algorithm 1 takes as input graph $G$ with $n$ nodes and $e$ edges, and outputs a line graph $\mathsf{L}(G')$ with $2e$ nodes.

## 2.3 UNITARY ADJACENCY MATRIX CALCULATION: COMPUTE THE EDGE WEIGHTS FOR MESSAGE PASSING

According to Proposition 2.2, the line graph $\mathsf{L}(G')$ has unitary adjacency matrices. In this section, we propose an algorithm to calculate a unitary adjacency matrix for GUMP, because the unitary adjacency matrix depends on the input graph and should be calculated for each graph.

### 2.3.1 PERMUTATION-EQUIVARIANT PROJECTION

Permutation equivariance of message passing is a key property for GNN to apply to graphs with varying node orders. To achieve this, the calculation of a unitary adjacency matrix has to be

permutation equivariance. Our method consists of two steps: 1) calculate edge weights to form a weighted adjacency matrix; 2) impose unitarity to the weighted adjacency matrix.

Firstly, edge weight for $(i, j) \in \mathsf{E}[\mathsf{L}(G')]$ is calculated with

$$\alpha_{ij} = \mathsf{Tanh}\left(\mathbf{w}^\top \cdot \mathsf{LeakyReLU}(\mathbf{W}_s\mathbf{h}_i + \mathbf{W}_t\mathbf{h}_j)\right), \tag{1}$$

where $\mathbf{h}_i$ ($\mathbf{h}_j$) is the representations for node $i$ ($j$) in $\mathsf{L}(G')$, $\mathbf{W}_s, \mathbf{W}_t \in \mathbb{R}^{d' \times d}$ are transformation matrices for source and target nodes of an edge respectively, and $\mathbf{w} \in \mathbb{R}^{d'}$ is a learnable parameter. Then, the weighted adjacency matrix of $\mathsf{L}(G')$, denoted as $\bar{\mathbf{A}} \in \mathbb{R}^{2e \times 2e}$, is formed from edge weights, i.e., $\bar{\mathbf{A}}_{ij} = \alpha_{ij}$.

After calculating $\bar{\mathbf{A}}$, we impose unitarity to $\bar{\mathbf{A}}$ by projection. We use the projection algorithm in Keller (1975), which takes advantage of the fact that the polar transformation yields the closest unitary matrix to a given matrix in terms of the Frobenius norm. The following lemma describes the unitary projection in GUMP:

**Lemma 2.3.** *Given a weighted adjacency matrix $\bar{\mathbf{A}} \in \mathbb{R}^{2e \times 2e}$ of $\mathsf{L}(G')$, the unitary projection of $\bar{\mathbf{A}}$ is given by $\mathsf{U}[\bar{\mathbf{A}}] := \arg\min_{\mathbf{U} \text{ is unitary}} \left\|\bar{\mathbf{A}} - \mathbf{U}\right\|_F^2 = \bar{\mathbf{A}}\left(\bar{\mathbf{A}}^\dagger\bar{\mathbf{A}}\right)^{-\frac{1}{2}}$.*

Lemma 2.3 (proved in Appendix D) indicates that $\mathsf{U}[\bar{\mathbf{A}}] = \bar{\mathbf{A}}\left(\bar{\mathbf{A}}^\dagger\bar{\mathbf{A}}\right)^{-\frac{1}{2}}$ is the unitary adjacency matrix for GUMP. Also, the unitary projection $\mathsf{U}[\bar{\mathbf{A}}]$ is guaranteed to be permutation-equivariant when $\bar{\mathbf{A}}$ is a full-rank matrix with the following proposition.

**Proposition 2.4** (Strong permutation equivariance)**.** *Given two permutation matrices $\mathbf{P}_1$ and $\mathbf{P}_2$, if $\bar{\mathbf{A}}$ is a full-rank matrix, the unitary projection $\mathsf{U}[\bar{\mathbf{A}}]$ is equivariant to both row and column permutations of $\bar{\mathbf{A}}$, i.e., $\mathbf{P}_1\mathsf{U}[\bar{\mathbf{A}}]\mathbf{P}_2^\top = \mathsf{U}[\mathbf{P}_1\bar{\mathbf{A}}\mathbf{P}_2^\top]$.*

By Proposition 2.4 (proved in Appendix D), the weighted adjacency matrix $\bar{\mathbf{A}}$ should be full-rank to guarantee permutation equivariance of GUMP. Empirically, inspired by GATv2 (Brody et al., 2021), $\bar{\mathbf{A}}$ induced by (1) is full-rank in experiments and thus can guarantee the permutation equivariance of GUMP. However, the unitary projection (Lemma 2.3) is computationally expensive due to the inverse square root of $\bar{\mathbf{A}}^\dagger\bar{\mathbf{A}} \in \mathbb{R}^{2e \times 2e}$.

### 2.3.2 FEASIBLE IMPLEMENTATION

By utilizing the intrinsic structure in the unitary adjacency matrix, the feasible unitary projection is implemented in Algorithm 2. In Algorithm 2, weighted adjacency matrix $\bar{\mathbf{A}}$ is first calculated with (1) in step 1. Then, the intrinsic structure of $\bar{\mathbf{A}}$ allows it to be permuted to be block diagonal with permutation matrices $\mathbf{P}_1$ and $\mathbf{P}_2$ in line 2. The permuted diagonal ma-

---

**Algorithm 2** Calculation of unitary adjacency matrix

**Require:** $\mathsf{L}(G')$ outputted by Algorithm 1;
1: Calculate $\bar{\mathbf{A}}$ of $\mathsf{L}(G')$ with (1);
2: Find the permutation matrices $\mathbf{P}_1, \mathbf{P}_2$ such that $\mathbf{D} :=$ $\mathrm{diag}(\mathbf{D}_1, \cdots, \mathbf{D}_b) = \mathbf{P}_1\bar{\mathbf{A}}\mathbf{P}_2^\top$ is block diagonal;
3: Calculate $\mathsf{U}[\mathbf{D}] = \mathrm{diag}(\mathsf{U}[\mathbf{D}_1], \mathsf{U}[\mathbf{D}_2], \cdots, \mathsf{U}[\mathbf{D}_b])$;
4: Calculate $\mathsf{U}[\bar{\mathbf{A}}] = \mathbf{P}_1^\top\mathsf{U}[\mathbf{D}]\mathbf{P}_2$;
5: **Return:** Unitary adjacency matrix $\mathsf{U}[\bar{\mathbf{A}}]$.

---

trix is denoted as $\mathbf{D}:=\mathrm{diag}(\mathbf{D}_1,\mathbf{D}_2,\cdots,\mathbf{D}_b)=\mathbf{P}_1\bar{\mathbf{A}}\mathbf{P}_2^\top$. By Proposition 2.4, the unitary projection of $\mathbf{D}$ is equal to the matrix after applying row permutation $\mathbf{P}_1$ and column permutation $\mathbf{P}_2$ to $\mathsf{U}[\bar{\mathbf{A}}]$, i.e., $\mathsf{U}[\mathbf{P}_1\bar{\mathbf{A}}\mathbf{P}_2^\top] = \mathbf{P}_1\mathsf{U}[\bar{\mathbf{A}}]\mathbf{P}_2^\top$, indicating $\mathsf{U}[\bar{\mathbf{A}}] = \mathbf{P}_1^\top\mathsf{U}[\mathbf{P}_1\bar{\mathbf{A}}\mathbf{P}_2^\top]\mathbf{P}_2$. Thus, $\mathsf{U}[\bar{\mathbf{A}}]$ can be efficiently calculated by first applying unitary projection to each block $\mathbf{D}_i$ of $\mathbf{D}$ in line 3 and then applying the inverse row and column permutation $\mathbf{P}_1^\top$ and $\mathbf{P}_2^\top$ to the unitary projection of $\mathbf{D}$ in line 4. The correctness of Algorithm 2 is guaranteed in Proposition 2.5 (proved in Appendix D) and Corollary 2.6 (proved in Appendix D) ensures the projected matrix $\mathsf{U}[\bar{\mathbf{A}}]$ has the same support as $\mathsf{L}(G')$.

**Proposition 2.5.** *The matrix returned by Algorithm 2 for graph $\mathsf{L}(G')$ is equal to $\bar{\mathbf{A}}\left(\bar{\mathbf{A}}^\dagger\bar{\mathbf{A}}\right)^{-\frac{1}{2}}$.*

**Corollary 2.6.** *With the strong permutation equivariance in Proposition 2.4, assuming each $\mathsf{U}[\mathbf{D}_i]$ is fully supported, the matrix returned by Algorithm 2 has the same support as the line graph $\mathsf{L}(G')$.*

Corollary 2.6 requires the unitary projection of each block $\mathbf{D}_i$ to be fully supported, which is not a strong assumption and empirically satisfied in experiments. Algorithm 2 is computationally

Table 1: Comparison of different methods for oversquashing on important properties of GNN and Jacobian measure. "Permutation equivariance" denotes the order of nodes in the graph does not affect the node representations of GNN. "Measure" represents the measure of oversquashing used in the corresponding method. "Jacobian measure w.r.t $L$" denotes the order of the expected Jacobian measure with respect to the number of layers $L$ (Theorems 2.1 and D.6 in Appendix D).

| Methods | SDRF | FoSR | LASER | GTR | GUMP |
|---|---|---|---|---|---|
| Permutation equivariance | ✓ | ✗ | ✓ | ✓ | ✓ |
| Graph connectivity | ✗ | ✗ | ✗ | ✗ | ✓ |
| Measure | curvature | spectral gap | walks | effective resistance | Jacobian |
| Jacobian measure w.r.t $L$ | $\mathcal{O}(c^L)$ | $\mathcal{O}(c^L)$ | $\mathcal{O}(c^L)$ | $\mathcal{O}(c^L)$ | $\mathcal{O}(1)$ |

feasible from two perspectives. Firstly, it applies unitary projection (Lemma 2.3) to block matrices $\mathbf{D}_i$, each with sizes $d_1, d_2, \cdots, d_b$ ($\sum_{i=1}^{b} d_i = 2e$). The computational complexity of Algorithm 2 is $\mathcal{O}(\sum_{i=1}^{b} d_i^3)$, in contrast to the complexity of $\mathcal{O}(de^3)$ when applied to the large matrix $\bar{\mathbf{A}}$. This results in lower computational cost for unitarity projection of block diagonal matrices, particularly when the sizes of the block matrices are small. Secondly, the algorithm benefits from the existence of many block matrices $\mathbf{D}_i$ with identical sizes because matrices of the same size can be grouped and computed in parallel with PyTorch.

---

**Algorithm 3** GNN with the graph unitary message passing mechanism (GNN-GUMP)

---

**Require:** A graph $G = (V, E, \mathbf{X})$;
1: $\mathbf{X}^{(0)} = \mathsf{GNN}(\mathbf{X}, G)$;
2: Transform $G$ to $\mathsf{L}(G')$ with Algorithm 1;
3: Generate initial representation $\mathbf{H}^{(0)}$ for $\mathsf{L}(G')$ with $\mathbf{h}_{(i,j)} = [\mathbf{x}_i^{(0)}; \mathbf{x}_j^{(0)}], \forall (i, j) \in \mathsf{V}[\mathsf{L}(G')]$;
4: Calculate $\mathsf{U}[\bar{\mathbf{A}}]$ with Algorithm 2;
5: **for** $k = 1 \cdots L$ **do**
6:     $\mathbf{h}_v^{(k)} = \gamma(\mathbf{h}_v^{(k-1)}, \sum_{u \in N_v} \mathsf{U}[\bar{\mathbf{A}}]_{vu} \phi^{(k)}(\mathbf{h}_v^{(k-1)}, \mathbf{h}_u^{(k-1)})), v \in V$
7: **end for**
8: Scatter $\mathbf{H}^{(L)}$ to nodes of $G$ with $\mathbf{H}_s^{(L)} = \mathsf{Scatter}(\mathbf{H}^{(L)}, G)$;
9: Generate node representations of $G$ with $\mathbf{X}^{(L)} = [\mathbf{X}^{(0)}; \mathbf{H}_s^{(L)}]$.
10: **Return:** Node representations $\mathbf{X}^{(L)}$ of $G$.

---

## 3 Apply GUMP to GNN

In this section, we apply GUMP to different GNN architectures for graph learning tasks in Algorithm 3. Given a graph $G$, a base GNN first computes the initial node representations of $G$, i.e., $\mathbf{X}^{(0)} = \mathsf{GNN}(\mathbf{X}, G)$. Then, Algorithm 1 transforms $G$ to $\mathsf{L}(G')$. The initial node representations $\mathbf{H}^{(0)} \in \mathbb{R}^{2e \times 2d}$ of $\mathsf{L}(G')$ are generated with $\mathbf{h}_{(i,j)} = [\mathbf{x}_i^{(0)}; \mathbf{x}_j^{(0)}], \forall (i, j) \in \mathsf{V}[\mathsf{L}(G')]$ ($i, j \in \mathsf{V}[G]$). Next, the unitary adjacency matrix $\mathsf{U}[\bar{\mathbf{A}}]$ of $\mathsf{L}(G')$ is calculated from Algorithm 2 and applied to propagate messages in graph with $\mathbf{h}_v^{(k)} = \gamma(\mathbf{h}_v^{(k-1)}, \sum_{u \in N_v} \mathsf{U}[\bar{\mathbf{A}}]_{vu} \phi^{(k)}(\mathbf{h}_v^{(k-1)}, \mathbf{h}_u^{(k-1)}))$, $v \in V$ with any graph convolution operator. After $L$ layers of unitary message passing, we obtain the node representations $\mathbf{H}^{(L)}$, which is later scattered to nodes of $G$ with $\mathbf{H}_s^{(L)} = \mathsf{Scatter}(\mathbf{H}^{(L)}, G) \in \mathbb{R}^{n \times d'}$. Then, $\mathbf{H}_s^{(L)}$ are concatenated with $\mathbf{X}^{(0)}$ to obtain the final node representations $\mathbf{X}^{(L)} = [\mathbf{X}^{(0)}; \mathbf{H}_s^{(L)}] \in \mathbb{R}^{n \times (d+d')}$ of $G$. Finally, various graph learning tasks, e.g., graph and node classification, link prediction, and graph regression, are performed based on $\mathbf{X}^{(L)}$. In this paper, GUMP is a general one-hop message-passing mechanism for GNN. Therefore, depending on the specific convolution operator in line 6 of Algorithm 3, GNN with GUMP is named as [GNN type]-GUMP in Section 4, e.g., GCN-GUMP and GIN-GUMP have graph convolution operator and graph isomorphism operator in line 6 of Algorithm 3, respectively.

### 3.1 Comparison with existing methods

GUMP paves a new way to solve the oversquashing problem instead of rewiring. Overall, GUMP has the following advantages: (1) GUMP is permutation-equivariant, which is a desirable property

for graph learning. (2) Unlike rewiring methods, GUMP does not introduce extra connectivity to the original graph and thus preserves the original graph connectivity. (3) GUMP achieves the optimal Jacobian measure of oversquashing since the eigenvalues of unitary adjacency matrices are complex units and thus will not change exponentially with respect to the number of GNN layers. The comparison of GUMP and other oversquashing methods are in Table 1.

Previous work on unitary GNN, e.g., Ortho-GConv (Guo et al., 2022), imposes unitarity on the feature transformation matrix of GNN. Unlike Ortho-GConv, GUMP addresses the issue of ill-posed gradient caused by oversquashing by imposing unitarity on the adjacency matrix. Moreover, enforcing unitarity on adjacency matrix is more challenging than that on feature transformation, since adjacency matrices depend on the input graph and are not parameters of GNN.

### 3.2 Positions of GUMP

In this paper, we focus exclusively on one-hop message passing, the fundamental mechanism in graph learning. To clarify our setting, we talk about the position of GUMP in graph learning from the following aspects.

**Multi-hop message passing**   We are aware of many multi-hop message passing methods (Feng et al., 2022), e.g, Drew (Gutteridge et al., 2023) and GRIT (Ma et al., 2023), which can alleviate the oversquashing problem, capture long-range interactions in graphs, and achieve better performance than GUMP in most datasets. We want to clarify that GUMP and multi-hop message passing methods are in orthogonal categories. GUMP focuses on improving the fundamental message passing mechanism, while multi-hop message passing methods applies multi-hop node information to improve the performance. GUMP addresses the oversquashing issue caused by fundamental message passing, and this issue also exists in multi-hop message passing methods. In future work, we will combine GUMP with multi-hop message passing for further performance improvement.

**Stable signal propagation**   Stable signal propagation (Poole et al., 2016; Schoenholz et al., 2022) is important for the scalability and robustness of deep neural network. The signal propagation is difficult to stabilize in GNNs because of the irregular data structure of graphs (Rong et al., 2019; Alon & Yahav, 2020). There are many works (Xu et al., 2018; Gasteiger et al., 2018) to improve the signal propagation in GNNs from model architecture perspective. From the data perspective, rewiring methods (Rong et al., 2019; Alon & Yahav, 2020) disrupt graph connectivity and do not fully address signal propagation issues. GUMP offers a comprehensive approach for stable signal propagation in GNNs, addressing instability from irregular graph data without losing graph connectivity. In the future, GUMP can inspire more data-perspective research on stable signal propagation in GNNs and help scale up GNNs.

## 4 Experiments

In this section, we perform experiments to evaluate GUMP on graph learning tasks. All experiments are implemented by PyTorch Geometric (Fey & Lenssen, 2019) and conducted on NVIDIA RTX 4090 GPUs and AMD EPYC 7763 CPUs.

### 4.1 Experiments on synthetic dataset

**Setup**   In this section, we conduct experiments on synthetic datasets, i.e., CrossedRing, Ring, and CliquePath, in Di Giovanni et al. (2023) to test GUMP. The performance is evaluated on the distances from source to target in the range of 4 to 28. In the experiments, we compare GCN-GUMP and GCN. The layer $L$ of GCN-GUMP and GCN is appropriately set up according to the distance $d$ between source and target in the synthetic datasets (i.e., $L = \lfloor d/2 \rfloor + 1$), such that the long-range interactions can be captured by GNN. We set the hidden dimension to be 32 for both GCN-GUMP and GCN. The hyperparameters of GCN-GUMP for synthetic datasets are in Table 5 of Appendix E.

**Results**   We plot the average results from three random seeds of GCN-GUMP and GCN experiments in Fig. 2. For two easier datasets, i.e., CrossedRing and Ring, GUMP achieves 100% accuracy when the distance ranges from 4 to 28. For the challenging CliquePath dataset, GCN-GUMP's performance

deteriorates to random guessing at a distance of 28. The results show that GUMP can help capture the long-range interactions in graph learning tasks. We compare with more baselines in Appendix E.

## 4.2 EXPERIMENTS ON THE TUDATASET

**Datasets** We select five graph datasets, i.e., Mutag, Proteins, Enzymes, NCI1, and NCI109 from the TUDataset (Morris et al., 2020). We chose these datasets because they consist of chemistry or biological graphs, where the atoms far apart may be closer in space, and long-distance propagation will have significant advantages. The statistics of these datasets are in Table 4 of Appendix E.

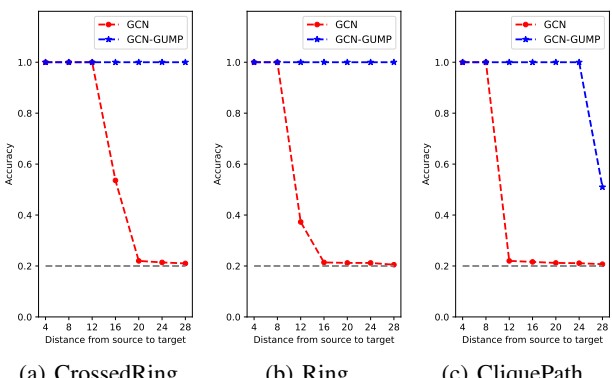

(a) CrossedRing     (b) Ring     (c) CliquePath

**Baselines** Because GUMP is a one-hop message-passing mechanism, we also compare one-hop message-passing baselines for fairness. Base-

Figure 2: The performance of GCN and GCN-GUMP on the CrossedRing, Ring, and CliquePath with different distances from source to target.

lines include various rewiring methods, i.e., DIGL (Gasteiger et al., 2019), SDRF (Topping et al., 2022), FoSR (Karhadkar et al., 2023), and GTR (Black et al., 2023). We use GCN and GIN as base GNN for comparison. The baselines follow the settings of Karhadkar et al. (2023). In Appendix E, we also compare GUMP with other methods for long-range graph learning and orthogonal GNN.

**Experimental details** To evaluate each method, we initially designate a test set comprising 10% of the graphs and a development set encompassing the remaining 90% of the graphs. The accuracies of each configuration are determined through 100 random train/validation splits of the development set, with 80% for training and 10% for validation. During the training phase, a stopping patience of 100 epochs is employed based on validation loss. Subsequently, for the test results, we report 95% confidence intervals for the best validation accuracy observed across the 100 runs.

The number of layers for rewiring methods is set to be in the range of one to five. The number of layers for GUMP is manually tuned because the long-range interactions in a graph can only be captured by increasing its layers. The detailed hyperparameters of GUMP for different datasets are presented in Table 5 of Appendix E.

**Results** The results of GUMP on the TUDataset are shown in Table 2. Firstly, the results show that GUMP outperforms all baselines on all datasets. In particular, GUMP outperforms baselines by a large margin on Mutag, Enzymes, NCI1, and NCI109. Also, GIN-GUMP achieves better performance than GCN-GUMP on all datasets, which indicates that graph convolution operations are crucial for performance. Moreover, since GUMP usually has more layers than baselines in these datasets, the experiments show that GCN and GIN with more layers have degraded performance on all datasets, showing that the improvement of GUMP does not come from increasing expressivity with more GNN layers.

## 4.3 EXPERIMENTS ON LRGB

In this section, we conduct experiments on the Long Range Graph Benchmark (LRGB) (Dwivedi et al., 2022), which is a set of GNN benchmarks involving long-range interactions. Two datasets are selected from LRGB for comparison, i.e., Peptides-func and Peptides-struct. The statistics of these datasets are shown in Table 4.

The experiments of LRGB are conducted following the standard settings in Dwivedi et al. (2022). We set SDRF, FoSR, GTR, LASER (Barbero et al., 2023), GRAND, and ADGN as baselines. The hyperparameters of GUMP and more results for LRGB are presented in Appendix E. All datasets are tested without any additional features, e.g., positional encoding. The results of LRGB are shown in Table 3. The results show that GUMP outperforms all baselines, which indicates that GUMP is more

Table 2: Graph classification accuracy on the TUDataset. **First**, second, and ~~third~~ best results are bold, underlined, and underwaved, respectively.

| Base GNN | Methods | Mutag | Proteins | Enzymes | NCI1 | NCI109 | Rank |
|---|---|---|---|---|---|---|---|
| GCN | None | 72.15±2.44 | 70.98±0.74 | 27.67±1.16 | 68.74±0.45 | 67.90±0.50 | 4.2 |
| | None (+layer) | 70.05±1.83 | 69.80±0.99 | 23.63±1.07 | 63.94±1.34 | 55.92±1.26 | 6.8 |
| | DIGL | 79.70±2.15 | 70.76±0.77 | 35.72±1.12 | 69.76±0.42 | 69.37±0.43 | 3.0 |
| | SDRF | 71.05±1.87 | 70.92±0.79 | 28.37±1.17 | 68.21±0.43 | 66.78±0.44 | 4.8 |
| | FoSR | 80.00±1.57 | 73.42±0.81 | 25.07±0.99 | 57.27±0.54 | 56.82±0.60 | 4.6 |
| | GTR | 79.10±1.86 | 72.59±2.48 | 27.52±0.99 | 69.37±0.38 | 67.97±0.47 | 3.6 |
| | GCN-GUMP | **84.89±1.63** | **74.88±0.87** | **36.02±1.43** | **77.97±0.42** | **75.85±0.44** | 1.0 |
| GIN | None | 77.70±3.60 | 70.80±0.83 | 33.80±1.12 | 75.65±0.49 | 74.93±0.46 | 4.0 |
| | None (+layer) | 69.80±2.75 | 68.71±0.96 | 25.92±1.07 | 73.49±0.46 | 72.47±0.53 | 6.6 |
| | DIGL | 79.80±2.08 | 70.71±0.67 | 35.74±1.20 | 79.37±0.43 | 76.88±0.39 | 2.8 |
| | SDRF | 78.40±2.80 | 69.81±0.79 | 35.82±1.09 | 74.55±0.54 | 73.89±0.43 | 4.2 |
| | FoSR | 78.00±2.22 | 75.11±0.82 | 29.20±1.38 | 70.15±0.47 | 69.93±0.45 | 5.2 |
| | GTR | 77.60±2.84 | 73.13±0.69 | 30.57±1.42 | 75.45±0.44 | 75.28±0.42 | 4.2 |
| | GIN-GUMP | **86.72±1.53** | 75.43±0.70 | **48.43±1.24** | **81.25±0.37** | **78.45±0.44** | 1.0 |

suitable for graph learning tasks involving long-range interactions than previous rewiring methods. The results of some rewiring methods, e.g., GTR, are worse than GCN, indicating that the greedy algorithm and measure used by it to alleviate oversquashing is not robust and may not help improve the performance of GNN in real applications.

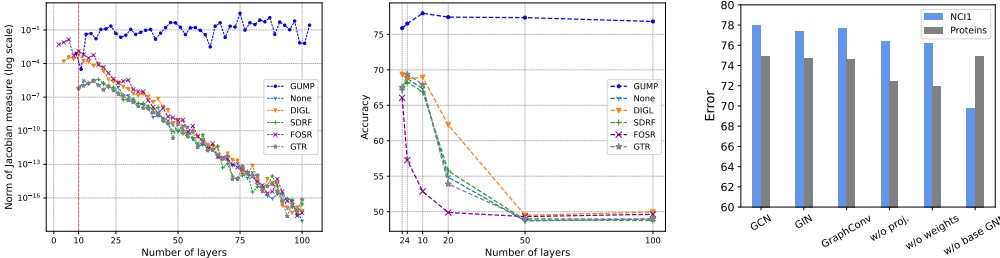

(a) Jacobian measure versus layers on NCI1

(b) Accuracy versus layers on NCI1

(c) Ablation on different components

Figure 3: Model analysis. GCN, GIN, and GUMP in (c) represent the convolution of GUMP. The base GNN of GUMP is GCN. "w/o proj" removes unitary projection in GUMP. "w/o weights" removes weighted adjacency matrix and unitary projection in GUMP. "w/o base GNN" removes base GNN in GUMP.

## 4.4 MODEL ANALYSIS

In this section, we perform more experiments to analyze GUMP from three aspects, i.e., Jacobian measure, number of layers, and ablation studies.

**Jacobian measure** We first visualize the Jacobian measure of oversquashing for GUMP and other baselines (more visualization in Appendix E). We choose a pair of nodes with a distance of ten from NCI1 and calculate the spectral norm of Jacobian measure for GUMP and baselines with base GNN as GCN. The visualization is shown in Fig. 3(a) with the norm of Jacobian measure in log scale. Firstly, when increas-

Table 3: Results of `Peptides-func` and `Peptides-struct`. **Bold** are best results.

| | Peptides-func Test AP ↑ | Peptides-struct Test MAE ↓ |
|---|---|---|
| GCN | .5930±.0023 | .3496±.0013 |
| SDRF | .5947±.0035 | .3404±.0015 |
| FoSR | .5947±.0027 | .3078±.0026 |
| GTR | .5075±.0029 | .3618±.0010 |
| LASER | .6440±.0010 | .3043±.0019 |
| GRAND | .5789±.0062 | .3418±.0015 |
| ADGN | .5975±.0044 | .2874±.0021 |
| GUMP | **.6843±.0037** | **.2467±.0021** |

ing the number of layers of GNNs, the Jacobian measure of GUMP does not decay, while the Jacobian measure of other baselines decays exponentially. The results validate the theoretical analysis in Theorems 2.1 and D.6, indicating that GUMP has the ability to capture long-range interactions in a graph without oversquashing. Secondly, the norm of the Jacobian measure varies for different baselines. For example, DIGL has a large norm of Jacobian measure when the number of layers is

smaller than 50. Therefore, DIGL performs better than other rewiring methods in Table 2. However, all baselines have a close norm of Jacobian measure when the number of layers is larger than 50.

**Deep GNN**  The number of GNN layers indicates the ability of GNNs to capture long-range interactions. We increase the number of layers of GUMP and baselines to see how their performances change. This experiment is conducted in NCI1 with base GNN as GCN and the number of layers in the range of 2 to 100. The other hyperparameters of GUMP are the same as Table 5. The results in Fig. 3(b) demonstrate that the performance of GUMP increases from 75.88% to 77.97% when increasing the number of layers from 2 to 10. However, the performance of baselines decreases a lot when increasing the number of layers. For example, the performance of FoSR decreases from 66.06% to 52.86% when the number of layers increases from 2 to 10. The performance of baselines changes drastically as the layers increase, while the performance of GUMP is more stable. The results show that GUMP can be deeper than previous methods, thus learning long-range interactions in graphs.

**Ablation studies**  Lastly, we conduct ablation studies to analyze GUMP in Fig. 3(c). We first replace the convolution of GUMP with GIN and GraphConv (Morris et al., 2019), showing that the choice of convolution can impact the performance of GUMP. Then, we remove unitary projection (i.e., message passing with $\bar{\mathbf{A}}$) and weighted adjacency matrix (i.e., message passing with $\bar{\mathbf{A}}[\mathsf{L}(G')]$) in GUMP and the results show that their performances decrease, indicating the importance of GUMP. Finally, we remove the base GNN in GUMP and the results show that the performance of GUMP varies on different datasets, i.e., the performance on NCI1 decreases, while the performance on Proteins does not decrease. This phenomenon is expected because the quality of the unitary adjacency matrix depends on the representations of nodes in the line graph (see (1)).

In model analysis, GUMP demonstrates the optimal Jacobian measure of oversquashing, and greater stability with increased layers compared to prior methods, highlighting the critical role of its design in achieving superior performance.

## 5  DISCUSSIONS AND LIMITATIONS

In this paper, we propose a novel method for oversquashing, i.e., Graph Unitary Message Passing (GUMP). Motivated by unitary RNNs, GUMP propagates messages on a graph with a unitary adjacency matrix. Compared to previous methods, GUMP achieves an optimal Jacobian measure of oversquashing, keeps the original graph connectivity, and is permutation-equivariant. We discuss below the limitations of GUMP and their implications for future work.

**Information loss**  Since GUMP involves graph transformation and adjacency matrix transformation, it is reasonable to consider what information is lost in GUMP. So we discuss the information loss in GUMP of different graphs: (1) *Undirected and unweighted graph:* These are the graphs discussed in our paper. For this graph, the information is not lost; (2) *Weighted graph:* For weighted graph, since GUMP utilizes a unitary adjacency matrix, the edge weights cannot be incorporated into message passing directly, and a feasible way for it is to convert edge weights to edge features in message passing, which is similar to R-GNN (Battaglia et al., 2018); (3) *Directed graph:* For directed graphs, the original directed edges become indistinguishable after the graph transformation in Appendix F, which means the original directionality is lost in the line graph. However, step 8 of Algorithm 3 can filter out the non-existing edges in the original graph, which can fix the information loss. For the unweighted digraph, we can follow the same procedure as the undirected and unweighted graph. So even though the information can be lost in weighted or directed graphs, there are other ways to incorporate the information into GNN when using GUMP.

**High computational cost**  The high cost of GUMP is attributed to the construction of the unitary projection. Although the unitary projection introduces significant computational complexity, it is essential for GUMP to obtain optimal Jacobian measure (as stated in Theorem 2.1) and exhibit good performance (as demonstrated in Fig. 3(c)). However, GUMP is a good option for many tasks, e.g., biology and chemistry, where accuracy is more important than time cost. Many graphs in these tasks are not very large, and the data collection and analysis processes often take significantly more time than model training. In the future, motivated by Orvieto et al. (2023), we will explore performing message passing in the diagonalized space of the adjacency matrix to reduce the computational cost of GUMP.

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

# A  THE ILLUSTRATED GUMP

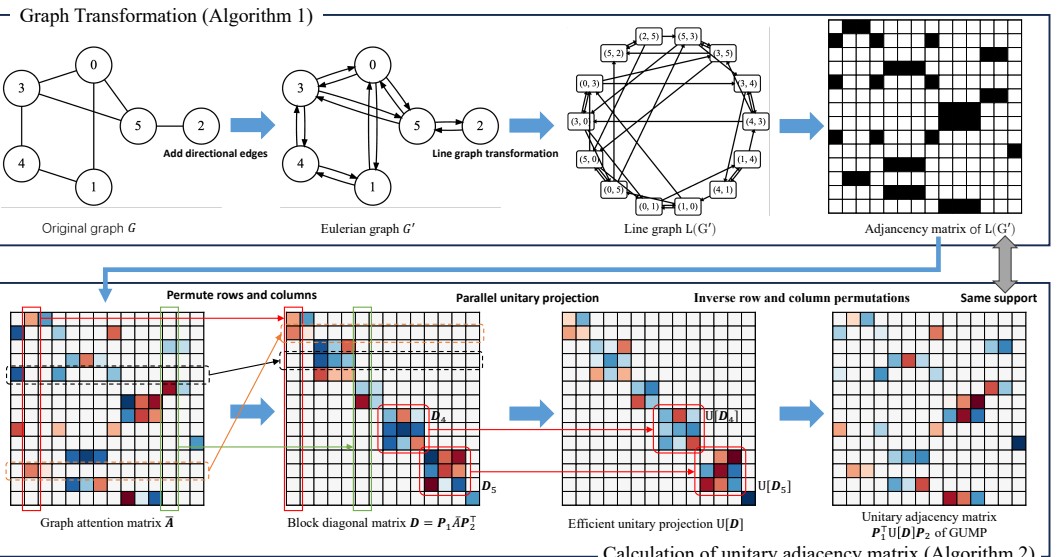

Figure 4: The illustrated GUMP.

# B  RELATED WORK

## B.1  GNN AND OVERSQUASHING

Graph neural network (GNN) (Kipf & Welling, 2016a; Gilmer et al., 2017) with $L$ layers is a type of neural network that uses graph $G$ and initial node features $\mathbf{H}^{(0)} = \mathbf{X}$ to learn node representations $\mathbf{H}^{(L)}$. The $k$-th layer of GNN updates node representation via the message-passing formula

$$\mathbf{h}_i^{(k)} = \delta(\mathbf{h}_i^{(k-1)}, \phi(\{\{\psi(\mathbf{h}_j^{(k-1)}), j \in \mathcal{N}(i)\}\})),$$

where $\delta$, $\phi$, and $\psi$ are combination, aggregation, and message functions respectively, $\{\{\cdots\}\}$ is a multiset, and $\mathcal{N}(i) = \{j | (i, j) \in E\}$.

Even though GNN achieves success in various graph learning tasks, it suffers from the oversquashing problem. The oversquashing problem is first noted by Alon & Yahav (2020). Inspired by Xu et al. (2018), Topping et al. (2022) proposes to measure oversquashing with the Jacobian between node features at different levels of a GNN. Based on the measure, Topping et al. (2022) propose a rewiring method to increase the curvature of the edges in a graph. Many works combat oversquashing by improving the spectral gap of a graph. Banerjee et al. (2022) measure oversquashing via the spectral gap of a graph, employing a rewiring algorithm based on expander graph construction and effective resistance for edge sampling. Karhadkar et al. (2023) introduce FoSR, a rewiring method that maximizes the first-order change in the spectral gap of the graph. Arnaiz-Rodríguez et al. (2022) introduced a GNN comprising a parameter-free layer for learning commute time and a rewiring layer to optimize spectral gap according to network characteristics and task requirements. Except from improving spectral gap, Black et al. (2023) focus on minimizing total resistance between node pairs and introduce the Greedy Total Resistance (GTR) rewiring method for oversquashing. Di Giovanni et al. (2023) analyze oversquashing with commute time. Barbero et al. (2023) propose a rewiring method to sequentially increase the number of walks between two nodes and preserve the locality in the original graph. Except for the one-hop message passing neural networks above, Gutteridge et al. (2023) propose a multi-hop message passing neural network (Drew) to alleviate oversquashing with layer-dependent graph rewiring and a delay mechanism for skip connections based on layer and node distance.

## B.2 Unitary neural networks

Unitary matrices $\mathbf{U}$ are square matrices satisfying $\mathbf{U}^\top \mathbf{U} = \mathbf{I}$. Unitary matrices contribute to the stability of neural network training by preserving vector norms and preventing issues like exploding or vanishing gradients. Imposing unitarity on neural networks enhances their ability to seamlessly capture and propagate information across layers. This technique has proven successful in various architectures, such as RNNs, CNNs, and GNNs.

Unitary neural networks (UNNs) are first developed to tackle the problem of vanishing and exploding gradients in RNNs, enabling more efficient learning of information in extremely long sequences of data compared to existing methods like LSTM (Hochreiter & Schmidhuber, 1997). To show how unitary matrices can alleviate the vanishing gradient problem in RNNs, we denote RNN as $\mathbf{h}_k = \sigma(\mathbf{W}\mathbf{h}_{k-1} + \mathbf{u}_k)$ with $\mathbf{h}_k$ as the hidden state at the $k$-th layer of RNN, $\mathbf{W}$ as the weight matrix, $\mathbf{u}_k$ as the $k$-th input of RNN, and $\sigma$ as the activation function. As shown in Fig. 1(a), the long-range dependency can be measured by $\partial \mathbf{h}_r / \partial \mathbf{u}_1$. To simplify the analysis, we assume the activation function $\sigma$ is identity. The Jacobian measure of the long-range dependency becomes $\partial \mathbf{h}_r / \partial \mathbf{u}_1 = \mathbf{W}^{r-1}$. When $\mathbf{W}$ is unitary, information in $\partial \mathbf{h}_r / \partial \mathbf{u}_1$ will not vanish or explode, which is the key to alleviate the vanishing/exploding gradient problem in learning long sequences.

Early algorithms on unitary RNN construct a series of parameterized unitary transformations to impose unitarity. EUNN (Jing et al., 2017) achieves this by composing layers of rotations, Fourier transforms, and other unitary transformations to parametrize unitary matrices. uRNN (Arjovsky et al., 2016) and scoRNN (Helfrich et al., 2018), on the other hand, maintain unitarity by performing a Cayley transformation to parametrize the full unitary space. expRNN (Lezcano-Casado & Martınez-Rubio, 2019), in contrast, parametrizes unitary matrices within the Lie algebra of the orthogonal/unitary group. projUNN (Kiani et al., 2022) first optimizes its parameters using gradient-based optimization and then maps the updated parameters to a unitary space. Most recently, linear recurrent unit (LRU) (Orvieto et al., 2023) significantly enhances the long-range learning capability of RNNs by linear RNN and initializing their weight matrix to be nearly unitary. Other works (Peng et al., 2023; Gu & Dao, 2023) on recurrent models also share the similar idea of LRU to effectively capture long-range dependencies in sequences, yielding impressive outcomes in language modeling and other tasks.

Despite RNN, unitarity has also been applied to CNNs and GNNs. Unitary CNNs (Sedghi et al., 2018; Li et al., 2019; Singla & Feizi, 2021; Trockman & Kolter, 2021) introduce various methods to restrict the convolutional filters to be unitary, e.g., via the Lie algebra of the orthogonal group (Singla & Feizi, 2021) and the Cayley transform (Trockman & Kolter, 2021). Ortho-GConv (Guo et al., 2022) imposes unitarity on the feature transformation matrix in GNNs.

## C  Preliminaries

**Definition C.1** (Line graph). Given a graph $G$, its line graph $\mathsf{L}(G)$ is a graph such that

- each vertex of $\mathsf{L}(G)$ represents an edge of $G$, i.e., $\mathsf{V}[\mathsf{L}(G)] = \mathsf{E}[G]$;

- two vertices of $\mathsf{L}(G)$ are adjacent if and only if their corresponding edges share a common endpoint in $G$, i.e., $\mathsf{E}[\mathsf{L}(G)] = \{((i,j),(j,k)) \in \mathsf{V}[\mathsf{L}(G)] \times \mathsf{V}[\mathsf{L}(G)] \mid (i,j),(j,k) \in \mathsf{E}[G]\}$.

**Definition C.2** (Eulerian graph). An Eulerian graph $G$ is a graph containing an Eulerian cycle, i.e., there is a trail in $G$ that starts and ends on the same vertex and visits every edge exactly once.

**Definition C.3** (Permutation matrix). A permutation matrix $\mathbf{P} \in \mathbb{R}^{n \times n}$ is a square binary matrix that has exactly one entry of 1 in each row and each column with all other entries 0.

**Theorem C.4.** *Every permutation matrix is orthogonal, i.e., if* $\mathbf{P}$ *is a permutation matrix,* $\mathbf{P}^\top \mathbf{P} = \mathbf{P}\mathbf{P}^\top = \mathbf{I}$.

# D   PROOF

## D.1   PROOF OF PROPOSITION 2.2

Proposition 2.2 is proved based on the following theorem and lemma. Theorem D.1 is a direct result from Theorem 3 in Severini (2003). Lemma D.2 is a well-known result in graph theory, which can be found in Theorem 1.7.2 of Bang-Jensen & Gutin (2008).

**Theorem D.1** (Existence of unitary adjacency matrix). *Let $G$ be a single-connected digraph. Its line graph $\mathsf{L}(G)$ (Definition C.1) is the digraph of a unitary matrix if and only if $G$ is Eulerian (Definition C.2).*

**Lemma D.2** (A special Eulerian graph). *A digraph graph is Eulerian if and only if it is connected and the in-degree and out-degree are equal at each vertex.*

*Proof of Proposition 2.2.* In Algorithm 1, the undirected edges in $G$ are split into two directed edges in $G'$. Therefore, the in-degree and out-degree of each vertex in $G'$ are equal, indicating $G'$ is Eulerian. Then, Theorem D.1 indicates that there exists unitary adjacency matrix $\mathbf{U}$ such that $\mathsf{S}[\mathbf{U}] = \mathbf{A}[\mathsf{L}(G')]$. $\square$

## D.2   PROOF OF LEMMA 2.3

*Proof.* Given any unitary $\mathbf{U}$, let $\mathbf{U} = \mathbf{M} + \mathsf{U}[\bar{\mathbf{A}}]$ for the properly chosen $\mathbf{M} \in \mathbb{C}^{2e \times 2e}$. Due to the unitarity of $\mathbf{U}$ and $\mathsf{U}[\bar{\mathbf{A}}]$, we have

$$\mathbf{M}\mathsf{U}[\bar{\mathbf{A}}]^\dagger + \mathbf{M}\mathbf{M}^\dagger + \mathsf{U}[\bar{\mathbf{A}}]\mathbf{M}^\dagger = 0. \tag{2}$$

Then, we have

$$\begin{aligned}
\|\bar{\mathbf{A}} - \mathbf{U}\|_F^2 &= \|\bar{\mathbf{A}} - \mathbf{M} - \mathsf{U}[\bar{\mathbf{A}}]\|_F^2 \\
&= \|\bar{\mathbf{A}} - \mathsf{U}[\bar{\mathbf{A}}]\|_F^2 + \mathrm{Tr}[\mathbf{M}\mathsf{U}[\bar{\mathbf{A}}]^\dagger + \mathbf{M}\mathbf{M}^\dagger + \mathsf{U}[\bar{\mathbf{A}}]\mathbf{M}^\dagger] - \mathrm{Tr}[\mathbf{M}^\dagger\bar{\mathbf{A}} + \bar{\mathbf{A}}^\dagger\mathbf{M}] \\
&= \|\bar{\mathbf{A}} - \mathsf{U}[\bar{\mathbf{A}}]\|_F^2 - \mathrm{Tr}[\mathbf{M}^\dagger\bar{\mathbf{A}} + \bar{\mathbf{A}}^\dagger\mathbf{M}] \\
&= \|\bar{\mathbf{A}} - \mathsf{U}[\bar{\mathbf{A}}]\|_F^2 - \mathrm{Tr}[\mathbf{M}^\dagger\mathsf{U}[\bar{\mathbf{A}}](\bar{\mathbf{A}}^\dagger\bar{\mathbf{A}})^{\frac{1}{2}} + (\bar{\mathbf{A}}^\dagger\bar{\mathbf{A}})^{\frac{1}{2}}\mathsf{U}[\bar{\mathbf{A}}]^\dagger\mathbf{M}].
\end{aligned}$$

Then, from (2), we have $\mathbf{M}\mathsf{U}[\bar{\mathbf{A}}]^\dagger + \mathsf{U}[\bar{\mathbf{A}}]\mathbf{M}^\dagger = -\mathbf{M}\mathbf{M}^\dagger$,

$$\|\bar{\mathbf{A}} - \mathbf{U}\|_F^2 = \|\bar{\mathbf{A}} - \mathsf{U}[\bar{\mathbf{A}}]\|_F^2 + \mathrm{Tr}[(\bar{\mathbf{A}}^\dagger\bar{\mathbf{A}})^{\frac{1}{2}}\mathbf{M}\mathbf{M}^\dagger]. \tag{3}$$

The second term above is non-negative because $\mathrm{Tr}[(\bar{\mathbf{A}}^\dagger\bar{\mathbf{A}})^{\frac{1}{2}}\mathbf{M}\mathbf{M}^\dagger] = \mathrm{Tr}[\mathbf{M}^\dagger(\bar{\mathbf{A}}^\dagger\bar{\mathbf{A}})^{\frac{1}{2}}\mathbf{M}]$ and $(\bar{\mathbf{A}}^\dagger\bar{\mathbf{A}})^{\frac{1}{2}}\mathbf{M}$ is positive semi-definite. Therefore, for all unitary $\mathbf{U}$,

$$\|\bar{\mathbf{A}} - \mathbf{U}\|_F^2 \geq \|\bar{\mathbf{A}} - \mathsf{U}[\bar{\mathbf{A}}]\|_F^2. \tag{4}$$

The result is proven. $\square$

## D.3   PROOF OF PROPOSITION 2.4

**Lemma D.3.** *For any unitary matrix $\mathbf{U}$, given two permutation matrices $\mathbf{P}_1$ and $\mathbf{P}_2$, $\mathbf{P}_1\mathbf{U}\mathbf{P}_2^\top$ is also unitary.*

*Proof.* Let $\hat{\mathbf{U}} = \mathbf{P}_1\mathbf{U}\mathbf{P}_2^\top$. Then, we have

$$\begin{aligned}
\hat{\mathbf{U}}\hat{\mathbf{U}}^\dagger &= \mathbf{P}_1\mathbf{U}\mathbf{P}_2^\top\mathbf{P}_2\mathbf{U}^\dagger\mathbf{P}_1^\top = \mathbf{P}_1\mathbf{U}\mathbf{U}^\dagger\mathbf{P}_1^\top = \mathbf{P}_1\mathbf{P}_1^\top = \mathbf{I}, \\
\hat{\mathbf{U}}^\dagger\hat{\mathbf{U}} &= \mathbf{P}_2\mathbf{U}^\dagger\mathbf{P}_1^\top\mathbf{P}_1\mathbf{U}\mathbf{P}_2^\top = \mathbf{P}_2\mathbf{U}^\dagger\mathbf{U}\mathbf{P}_2^\top = \mathbf{P}_2\mathbf{P}_2^\top = \mathbf{I}
\end{aligned}$$

which proves $\mathbf{P}_1\mathbf{U}\mathbf{P}_2^\top$ is unitary. $\square$

*Proof of Proposition 2.4.* This proposition is proved by the uniqueness of the unitary matrix $\mathsf{U}[\bar{\mathbf{A}}]$ when $\bar{\mathbf{A}}$ is a full-rank matrix.

Assume there exists another unitary matrix $\mathbf{U} = \mathbf{M} + \mathsf{U}[\bar{\mathbf{A}}]$ such that

$$\mathbf{U} = \arg\min_{\mathbf{U} \text{ is unitary}} \left\| \bar{\mathbf{A}} - \mathbf{U} \right\|_F^2. \tag{5}$$

According to the proof of Lemma 2.3, we have

$$\|\bar{\mathbf{A}} - \mathbf{U}\|_F^2 = \|\bar{\mathbf{A}} - \mathsf{U}[\bar{\mathbf{A}}]\|_F^2 + \text{Tr}[(\bar{\mathbf{A}}^\dagger \bar{\mathbf{A}})^{\frac{1}{2}} \mathbf{M}\mathbf{M}^\dagger].$$

Since $\mathbf{A}$ is full-rank matrix, $(\bar{\mathbf{A}}^\dagger \bar{\mathbf{A}})^{\frac{1}{2}}$ is position definite. Therefore, we have

$$\|\bar{\mathbf{A}} - \mathbf{U}\|_F^2 > \|\bar{\mathbf{A}} - \mathsf{U}[\bar{\mathbf{A}}]\|_F^2. \tag{6}$$

which contradicts the assumption that $\mathbf{U}$ is the minimizer of $\left\| \bar{\mathbf{A}} - \mathbf{U} \right\|_F^2$. Thus, the unitary projection of $\bar{\mathbf{A}}$ is unique when $\bar{\mathbf{A}}$ is full-rank.

Because $\mathsf{U}[\bar{\mathbf{A}}]$ is unique and is the minimizer of $\left\| \bar{\mathbf{A}} - \mathbf{U} \right\|_F^2$, from Lemma D.3, we have $\mathbf{P}_1 \mathsf{U}[\bar{\mathbf{A}}]\mathbf{P}_2^\top$ the minimizer of

$$\arg\min_{\mathbf{U} \text{ is unitary}} \left\| \mathbf{P}_1 \bar{\mathbf{A}} \mathbf{P}_2^\top - \mathbf{U} \right\|_F^2$$

for any permutation matrices $\mathbf{P}_1$ and $\mathbf{P}_2$. Because $\mathbf{P}_1 \bar{\mathbf{A}} \mathbf{P}_2^\top$ is also full-rank, $\mathbf{P}_1 \mathsf{U}[\bar{\mathbf{A}}]\mathbf{P}_2^\top$ is the unitary projection of $\mathbf{P}_1 \bar{\mathbf{A}} \mathbf{P}_2^\top$, which proves that $\mathsf{U}[\mathbf{P}_1 \bar{\mathbf{A}} \mathbf{P}_2^\top] = \mathbf{P}_1 \mathsf{U}[\bar{\mathbf{A}}]\mathbf{P}_2^\top$ for any permutation matrices $\mathbf{P}_1$ and $\mathbf{P}_2$. $\qquad\square$

### D.4 PROOF OF PROPOSITION 2.5

We need the following lemma to prove Proposition 2.5.

**Lemma D.4.** *Given the adjacency matrix* $\mathbf{A}[\mathsf{L}(G')]$, *its rows and columns can be permuted to transform* $\mathbf{A}[\mathsf{L}(G')]$ *to be block diagonal.*

*Proof.* According to Theorem 2 in Severini (2003), since $\mathsf{L}(G')$ is line graph, $\mathsf{L}(G')$ is specular. By Lemma 1 in Severini (2003), since $\mathsf{L}(G')$ is the digraph of a unitary matrix, $\mathsf{L}(G')$ is strongly quadrangular. Then, by Theorem 1 in Severini (2003), since $\mathsf{L}(G')$ is specular and strongly quadrangular, $\mathbf{A}[\mathsf{L}(G')]$ is composed of independent matrices, thus its rows and columns can be permuted to transform $\mathbf{A}[\mathsf{L}(G')]$ to be block diagonal. $\qquad\square$

*Proof of Proposition 2.5.* This proposition is proved by the following step-by-step analysis.

1. By Lemma D.4, we can permute the rows and columns of $\mathbf{A}[\mathsf{L}(G')]$ to transform $\mathbf{A}[\mathsf{L}(G')]$ to be block diagonal, i.e., $\mathbf{D} := \text{diag}(\mathbf{D}_1, \cdots, \mathbf{D}_b) = \mathbf{P}_1 \mathbf{A}[\mathsf{L}(G')]\mathbf{P}_2^\top$ is block diagonal, where $\mathbf{P}_1$ and $\mathbf{P}_2$ are permutation matrices.

2. Since $\bar{\mathbf{A}}$ is full-rank, by Proposition 2.4, the unitary projection of $\mathbf{D}$ is equal to the matrix after applying row permutation $\mathbf{P}_1$ and column permutation $\mathbf{P}_2$ to $\mathsf{U}[\bar{\mathbf{A}}]$, i.e., $\mathsf{U}[\mathbf{P}_1 \bar{\mathbf{A}} \mathbf{P}_2^\top] = \mathbf{P}_1 \mathsf{U}[\bar{\mathbf{A}}]\mathbf{P}_2^\top$. Using the property of permutation matrix (Theorem C.4), we have $\mathsf{U}[\bar{\mathbf{A}}] = \mathbf{P}_1^\top \mathsf{U}[\mathbf{P}_1 \bar{\mathbf{A}} \mathbf{P}_2^\top]\mathbf{P}_2$.

3. Finally, we have

$$\begin{aligned}
\mathsf{U}[\bar{\mathbf{A}}] &= \mathbf{P}_1^\top \mathsf{U}[\mathbf{P}_1 \bar{\mathbf{A}} \mathbf{P}_2^\top]\mathbf{P}_2 \\
&= \mathbf{P}_1^\top \mathsf{U}[\mathbf{D}]\mathbf{P}_2 \\
&= \mathbf{P}_1^\top \text{diag}(\mathsf{U}[\mathbf{D}_1], \cdots, \mathsf{U}[\mathbf{D}_b])\mathbf{P}_2,
\end{aligned}$$

which proves the correctness of Algorithm 2.

$\qquad\square$

### D.5 PROOF OF COROLLARY 2.6

*Proof.* From Algorithm 2, the support of the line graph $\mathsf{L}(G')$ can be permuted by rows and columns to be a block diagonal matrix $\mathbf{D} = \mathrm{diag}\{\mathbf{D_1}, \cdots, \mathbf{D_b}\}$ with each block $\mathbf{D}'_i$ a dense support, i.e., $\mathbf{D_i} = \mathbf{1}$.

Then, for each dense sub-matrix $\mathbf{D}_i$ from the block diagonal matrix, there are unitary matrices with support $\mathbf{D}_i$. Therefore, there are unitary matrices, i.e., $\mathsf{U}[\mathbf{P}_1 \bar{\mathbf{A}} \mathbf{P}_2^\top]$ in our paper, with support $\mathbf{D}$.

Since row and column permutations are invertible, we can permute the support of the unitary matrix back to the original support of the line graph $\mathsf{L}(G')$, i.e., $\mathbf{P}_1^\top \mathsf{U}[\mathbf{P}_1 \bar{\mathbf{A}} \mathbf{P}_2^\top] \mathbf{P}_2$ has the same support as $\mathbf{D}$. Because the permutation matrices $\mathbf{P}_1$ and $\mathbf{P}_2$ are unitary, the matrix $\mathbf{P}_1^\top \mathsf{U}[\mathbf{P}_1 \bar{\mathbf{A}} \mathbf{P}_2^\top] \mathbf{P}_2$ is also a unitary matrix. 4. In Proposition 2.4, we show that $\mathsf{U}[\bar{\mathbf{A}}] = \mathbf{P}_1^\top \mathsf{U}[\mathbf{P}_1 \bar{\mathbf{A}} \mathbf{P}_2^\top] \mathbf{P}_2$. Thus, $\mathsf{U}[\bar{\mathbf{A}}]$ has the same support as the line graph $\mathsf{L}(G')$. $\qquad\square$

### D.6 PROOF OF THEOREM 2.1

Our proof is based on the GNN model from Fig. 1(a) with the activation function being ReLU. GUMP is analyzed with $\mathbf{A}$ being unitary and classical message passing is analyzed with $\mathbf{A}$ being the normalized adjacency matrix. Motivated by Xu et al. (2018), we analyze with the expected Jacobian measure.

Motivated by Xu et al. (2018), we first introduce the expected Jacobian measure of oversquashing.

**Theorem D.5.** *Given a $L$-layer GNN with ReLU as activation function, i.e., $\mathbf{H}^{(k)} = \mathrm{ReLU}(\mathbf{A}\mathbf{H}^{(k-1)}\mathbf{W}_k), \mathbf{H}^{(0)} = \mathbf{X}, k = 1 \cdots L$, assume that all paths in the computation graph of the model are activated with the same probability of success $\rho$, the expected Jacobian measure of oversquashing is*

$$\mathbb{E}\left[\frac{\partial \mathbf{h}_i^{(L)}}{\partial \mathbf{x}_s}\right] = \rho \prod_{l=L}^{1} \mathbf{W}_l^\top \left(\mathbf{A}^L\right)_{is}, \qquad (7)$$

*Proof.* Denote by $\mathbf{f}_i^{(l)}$ the pre-activated feature of $\mathbf{h}_i^{(l)}$, i.e., $\mathbf{f}_i^{(l)} = \sum_{z \in \mathcal{N}(i)} \mathbf{A}_{iz} \mathbf{h}_z^{(l-1)} \mathbf{W}_l$, for any $l = 1 \cdots L$, we have

$$\frac{\partial \mathbf{h}_i^{(l)}}{\partial \mathbf{h}_s^{(0)}} = \mathrm{diag}\left(\mathbb{1}_{\mathbf{f}_i^{(l)}>0}\right) \cdot \left(\sum_{z \in \mathcal{N}(i)} \mathbf{A}_{iz} \frac{\partial \mathbf{h}_z^{(l-1)}}{\partial \mathbf{h}_s^{(0)}}\right) \cdot \mathbf{W}_l^\top.$$

By the chain rule, we get

$$\frac{\partial \mathbf{h}_i^{(L)}}{\partial \mathbf{h}_s^{(0)}} = \sum_{p=1}^{\Psi} \left[\frac{\partial \mathbf{h}_i^{(L)}}{\partial \mathbf{h}_s^{(0)}}\right]_p$$

$$= \sum_{p=1}^{\Psi} \prod_{l=L}^{1} \mathrm{diag}\left(\mathbb{1}_{\mathbf{f}_{v_p^l}^{(l)}>0}\right) \mathbf{A}_{v_p^l v_p^{l-1}} \mathbf{W}_l^\top.$$

Here, $\Psi$ is the total number of paths $v_p^L v_p^{L-1} \cdots v_p^1 v_p^0$ of length $L+1$ from $v_p^0 = s$ to $v_p^L = i$. For $l = 1 \cdots L - 1, v_p^{l-1} \in \mathcal{N}(v_p^l)$.

For each path $p$, the derivative $[\partial \mathbf{h}_i^{(L)}/\partial \mathbf{h}_s^{(0)}]_p$ represents a directed acyclic computation graph. At a layer $l$, we can express an entry of the derivative as

$$\left[\frac{\partial \mathbf{h}_i^{(L)}}{\partial \mathbf{h}_s^{(0)}}\right]_p^{(m,n)} = \prod_{l=L}^{1} \mathbf{A}_{v_p^l v_p^{l-1}} \sum_{q=1}^{\Phi} Z_q \prod_{l=L}^{1} w_q^{(l)},$$

where $\Phi$ is the number of paths $q$ from the input neurons to the output neuron $(m, n)$, in the computation graph of $[\partial \mathbf{h}_i^{(L)}/\partial \mathbf{h}_s^{(0)}]_p$. For each layer $l$, $w_q^{(l)}$ is the entry of $\mathbf{W}_l^\top$ that is used in the

$q$-th path. Finally, $Z_q \in \{0, 1\}$ represents whether the $q$-th path is active ($Z_q = 1$) or not ($Z_q = 0$) as a result of ReLU activation of the entries of $\mathbf{f}_{v_p^l}^{(l)}$'s on the $q$-th path.

Under the assumption that $Z_q$ is a Bernoulli random variable with success probability $\rho$. Because of $\mathbb{P}[Z_q = 1] = \rho, \forall q$, we have

$$\mathbb{E}\left[\left[\frac{\partial \mathbf{h}_i^{(L)}}{\partial \mathbf{h}_s^{(0)}}\right]_p^{(m,n)}\right] = \rho \prod_{l=L}^1 \mathbf{A}_{v_p^l v_p^{l-1}} \sum_{q=1}^\Phi \prod_{l=L}^1 w_q^{(l)}.$$

Then, the expected Jacobian measure of oversquashing is

$$\mathbb{E}\left[\frac{\partial \mathbf{h}_i^{(L)}}{\partial \mathbf{x}_s}\right] = \sum_{p=1}^\Psi \mathbb{E}\left[\left[\frac{\partial \mathbf{h}_i^{(L)}}{\partial \mathbf{h}_s^{(0)}}\right]_p\right] = \rho \prod_{l=L}^1 \mathbf{W}_l^\top \left(\mathbf{A}^L\right)_{is}.$$

$\square$

*Proof of Theorem 2.1.* We analyze different components in the expected Jacobian measure (7).

Firstly, $\prod_{l=L}^1 \mathbf{W}_l^\top$ is the product of weight matrices in GNNs, which will not change exponentially with respect to $L$ because the weight matrices are appropriately initialized to avoid exploding or decay.

Then, we focus the analysis on $\mathbf{A}^L$. We first diagonalize $\mathbf{A}$ in complex space, i.e., $\mathbf{A} = \mathbf{P}\mathbf{\Lambda}\mathbf{P}^{-1}$ with $\mathbf{\Lambda}$ being a diagonal matrix with its diagonal elements being eigenvalues of $\mathbf{A}$. Since $\mathbf{A}$ is unitary, the elements in $\mathbf{\Lambda}$ are complex units, i.e., $\mathbf{\Lambda} = \text{diag}(e^{i\theta_1}, e^{i\theta_2}, \cdots, e^{i\theta_{2e}})$. Therefore, $\mathbf{A}^L$ is equal to $\mathbf{P}\mathbf{\Lambda}^L\mathbf{P}^{-1}$, where $\Lambda = \text{diag}(e^{i\theta_1 L}, e^{i\theta_2 L}, \cdots, e^{i\theta_{2e} L})$ which is also a diagonal matrix with its diagonal elements being complex units. Then, we have $(\mathbf{A}^L)_{is} = \mathbf{P}_i \text{diag}(e^{i\theta_1 L}, e^{i\theta_2 L}, \cdots, e^{i\theta_{2e} L})(\mathbf{P}^{-1})_s$, which is a value that does not change exponentially with respect to $L$ and can be bounded by constants. With Euler's formula, i.e., $e^{ix} = \cos x + i \sin x$, the relation between $(\mathbf{A}^L)_{is}$ and $L$ is a trigonometric function.

Since the trigonometric function can be bounded by constants, we have $\prod_{l=L}^1 \mathbf{W}_l^\top \left(\mathbf{A}^L\right)_{is}$ is bounded by constants, i.e., $\mathbb{E}\left[\partial \mathbf{h}_i^{(L)}/\partial \mathbf{x}_s\right] = \mathcal{O}(1)$. $\square$

### D.7 THEORY OF CLASSICAL MESSAGE PASSING

The next theorem analyzes the expected Jacobian measure of classical message passing.

**Theorem D.6.** *The expected Jacobian measure for classical message passing, i.e., $\mathbf{A} = \hat{\mathbf{A}}$ in GNN, is approximately in the order of $\mathbb{E}\left[\partial \mathbf{h}_i^{(L)}/\partial \mathbf{x}_s\right] = \mathcal{O}(c^L)$, where $c \in (0, 1)$.*

Theorem D.6 shows that the Jacobian measure of classical message passing decays exponentially concerning $L$, thus leading to oversquashing. Therefore, Theorems 2.1 and D.6 indicate that GUMP can achieve an optimal Jacobian measure of oversquashing, i.e., $\mathcal{O}(1)$, while the classical message passing cannot. Theorems 2.1 and D.6 are validated by experiments in Section 4.4.

*proof of Theorem D.6.* We analyze the different components in the expected Jacobian measure (7).

Firstly, $\prod_{l=L}^1 \mathbf{W}_l^\top$ is the product of weight matrices in GNNs, which will not change exponentially with respect to $L$ because the weight matrices are appropriately initialized to avoid exploding or decay.

Then, we focus the analysis on $\mathbf{A}^L$. We first diagonalize $\mathbf{A}$ in complex space, i.e., $\mathbf{A} = \mathbf{P}\mathbf{\Lambda}\mathbf{P}^{-1}$ with $\mathbf{\Lambda}$ being a diagonal matrix with its diagonal elements being eigenvalues of $\mathbf{A}$. Since $\mathbf{A} = \hat{\mathbf{A}}$, the elements in $\mathbf{\Lambda}$ are in the range of 0 and 1, i.e., $\mathbf{\Lambda} = \text{diag}(\theta_1, \theta_2, \cdots, \theta_{2e}), \theta_i \in [0, 1]$. Therefore, $\mathbf{A}^L$ is equal to $\mathbf{P}\mathbf{\Lambda}^L\mathbf{P}^{-1}$, where $\Lambda = \text{diag}(\theta_1^L, \theta_2^L, \cdots, \theta_{2e}^L)$. Without any assumption on the graph structure, there are many eigenvalues smaller than one. Then, we have $(\mathbf{A}^L)_{is} = \mathbf{P}_i \text{diag}(\theta_1^L, \theta_2^L, \cdots, \theta_{2e}^L)(\mathbf{P}^{-1})_s$, which is a value that change exponentially with respect to $L$.

Finally, we have $\prod_{l=L}^1 \mathbf{W}_l^\top \left(\mathbf{A}^L\right)_{is}$ in the order of $\mathcal{O}(c^L)$, i.e., $\mathbb{E}\left[\partial \mathbf{h}_i^{(L)}/\partial \mathbf{x}_s\right] = \mathcal{O}(c^L), c \in (0, 1)$. $\square$

## E    MORE EXPERIMENTAL RESULTS

**Statistics of datasets**    The statistics of datasets used in experiments are shown in Table 4.

Table 4: Statistics of datasets.

|  | #graphs | Avg. nodes | Avg. edges | Task type |
|---|---|---|---|---|
| Mutag | 188 | 17.9 | 39.6 | Graph Classification |
| Proteins | 1,113 | 39.1 | 145.6 | Graph Classification |
| Enzymes | 600 | 32.6 | 124.3 | Graph Classification |
| NC1 | 4110 | 29.87 | 32.30 | Graph Classification |
| NC109 | 4127 | 29.68 | 32.13 | Graph Classification |
| Peptides-func | 15,535 | 150.94 | 307.30 | Graph Classification |
| Peptides-struct | 15,535 | 150.94 | 307.30 | Graph Regression |

**Hyperparameters of GUMP**    The hyperparameters of GUMP for both synthetic and real datasets are shown in Table 5.

Table 5: Hyperparameters of GUMP for datasets in experiments. $\text{layer}_{\text{GUMP}}$, $\text{lr}_{\text{base}}$, $\text{wd}_{\text{base}}$, $\text{lr}_{\text{GUMP}}$, $\text{wd}_{\text{GUMP}}$, drop., $d'$, $d$, batch size, $\text{layer}_{\text{base}}$, opt., sched., and epoch denotes the number of layers of GUMP, the learning rate of base GNN, weight decay of base GNN, the learning rate of GUMP, weight decay of GUMP, dropout rate, dimension of calculating (1), hidden dimension of GNN, batch size, number of layers of base GNN, optimizer, scheduler, and number of epochs, respectively.

|  | $\text{layer}_{\text{GUMP}}$ | $\text{lr}_{\text{base}}$ | $\text{wd}_{\text{base}}$ | $\text{lr}_{\text{GUMP}}$ | $\text{wd}_{\text{GUMP}}$ | drop. | $d'$ | $d$ | batch size | $\text{layer}_{\text{base}}$ | opt. | sched. | epoch |
|---|---|---|---|---|---|---|---|---|---|---|---|---|---|
| CrossedRing | - | $10^{-4}$ | $10^{-6}$ | $10^{-4}$ | 0 | 0 | 32 | 32 | 20 | 0 | adam | none | 200 |
| Ring | - | $10^{-4}$ | $10^{-6}$ | $10^{-4}$ | 0 | 0 | 32 | 32 | 20 | 0 | adam | none | 200 |
| CliquePath | - | $10^{-4}$ | $10^{-6}$ | $10^{-4}$ | 0 | 0 | 32 | 32 | 20 | 0 | adam | none | 200 |
| Mutag | 16 | $10^{-2}$ | $10^{-4}$ | $10^{-4}$ | 0 | 0 | 32 | 64 | 16 | 5 | adam | none | 100 |
| Proteins | 20 | $10^{-2}$ | $10^{-2}$ | $10^{-4}$ | $10^{-2}$ | 0 | 32 | 64 | 64 | 3 | adam | none | 100 |
| Enzymes | 10 | $10^{-2}$ | $10^{-4}$ | $10^{-4}$ | 0 | 0 | 32 | 64 | 16 | 1 | adam | none | 100 |
| NC1 | 10 | $10^{-2}$ | $10^{-4}$ | $10^{-4}$ | 0 | 0 | 32 | 64 | 16 | 1 | adam | none | 100 |
| NC109 | 10 | $10^{-2}$ | $10^{-4}$ | $10^{-4}$ | 0 | 0 | 32 | 64 | 16 | 1 | adam | none | 100 |
| Peptides-func | 12 | 0.005 | 0.1 | 0.1 | 0.1 | 0.2 | 32 | 256 | 200 | 3 | adam | cos. | 250 |
| Peptides-struct | 12 | 0.005 | 0.1 | 0.005 | 0.1 | 0.2 | 32 | 256 | 200 | 3 | adam | cos. | 250 |

**Comparison GUMP with more methods**    For the synthetic datasets, we compare GUMP with Drew and ADGN on synthetic datasets in Fig. 5. The results show that the performances of GUMP and Drew are close, while ADGN performs worse.

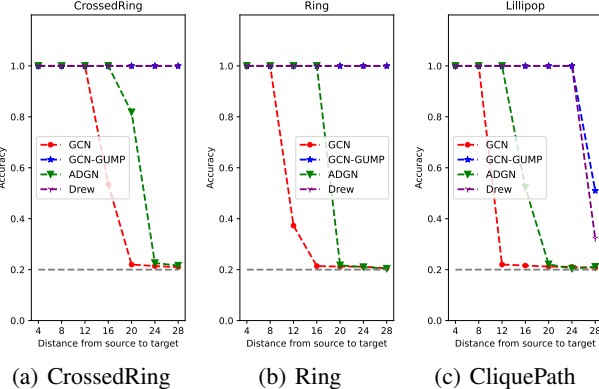

(a) CrossedRing     (b) Ring     (c) CliquePath

Figure 5: The performance of GCN and GCN-GUMP on the CrossedRing, Ring, and CliquePath with different distances from source to target.

We further compare the performances of GUMP, rewiring methods, graph neural diffusion, Graph Transformer, and orthogonal GNN on the TUDataset in Table 6. The compared methods are ADGN (Gravina et al., 2022), GRAND (Chamberlain et al., 2021)), Graph Transformer (Yun et al., 2019), and Ortho-GConv (Guo et al., 2022).

Table 6: Graph classification accuracy on the TUDataset. **First**, second, and third best results are bold, underlined, and underwaved, respectively.

| Classes | Methods | Mutag | Proteins | Enzymes | NCI1 | NCI109 |
|---|---|---|---|---|---|---|
| – | GCN | 72.15±2.44 | 70.98±0.74 | 27.67±1.16 | 68.74±0.45 | 67.90±0.50 |
| | GCN (+layer) | 70.05±1.83 | 69.80±0.99 | 23.63±1.07 | 63.94±1.34 | 55.92±1.26 |
| Rewiring (GCN) | DIGL | 79.70±2.15 | 70.76±0.77 | 35.72±1.12 | 69.76±0.42 | 69.37±0.43 |
| | SDRF | 71.05±1.87 | 70.92±0.79 | 28.37±1.17 | 68.21±0.43 | 66.78±0.44 |
| | FoSR | 80.00±1.57 | 73.42±0.81 | 25.07±0.99 | 57.27±0.54 | 56.82±0.60 |
| | GTR | 79.10±1.86 | 72.59±2.48 | 27.52±0.99 | 69.37±0.38 | 67.97±0.47 |
| Diffusion | ADGN | 81.39±1.81 | 73.81±0.80 | 28.78±1.25 | 76.15±0.42 | 74.31±0.44 |
| | GRAND | 77.94±1.73 | 73.24±0.94 | 24.13±1.05 | 68.51±0.48 | 67.26±0.46 |
| Transformer | Transformer | 69.15±1.78 | 66.21±0.96 | 28.33±1.44 | 58.41±0.55 | 58.25±0.52 |
| Ortho-GNN | Ortho-GConv | 71.78±2.52 | 63.80±0.98 | 18.30±1.13 | 69.92±0.60 | 68.91±0.50 |
| Ours | GCN-GUMP | **84.89±1.63** | **74.88±0.87** | **36.02±1.43** | **77.97±0.42** | **75.85±0.44** |

For the sake of fair comparison and experimental integrity, we also modified the multi-hop message passing methods (i.e., Drew and GRIT) into the one-hop variants. We replace $\sum_{k=1}^{\ell+1} \sum_{j \in \mathcal{N}_k(i)}$ in DRew with $\sum_{j \in \mathcal{N}_1(i)}$ and set $K$ of RRWP in GRIT to 2, making Drew and GRIT one-hop message passing methods. We report their results in Table 7. The results show that GUMP outperforms Drew and GRIT in one-hop message passing setting.

Table 7: Comparison with Drew and GRIT by setting their message-passing hop to one.

| | Peptides-func | Peptides-struct | Mutag | Proteins | Enzymes | NCI1 | NCI109 |
|---|---|---|---|---|---|---|---|
| Drew (1-hop) | 0.6996±0.0076 | 0.2881±0.0024 | 79.91±1.97 | 74.12±0.90 | 35.02±1.22 | 73.58±0.41 | 72.27±0.49 |
| GRIT (1-hop) | 0.6779±0.0079 | 0.2671±0.0018 | 80.76±2.18 | 73.71±0.89 | 35.22±1.17 | 72.21±0.46 | 71.68±0.44 |
| GCN-GUMP | 0.6843±0.0037 | 0.2467±0.0021 | 84.89±1.63 | 74.88±0.87 | 36.02±1.43 | 77.97±0.42 | 75.85±0.44 |

We also compare the Jacobian of GUMP with Drew and ADGN in Fig. 6. Jacobian of Drew exponentially increases, suggesting its potential numerical instability when training Drew with more layers. The Jacobian of ADGN is small when the ADGN layer is small and steadily increases to $10^{-8}$ as the ADGN layer reaches 100. Although the Jacobian of ADGN does not exhibit exponential decay, the correlation between distant nodes is significantly weaker compared to GUMP.

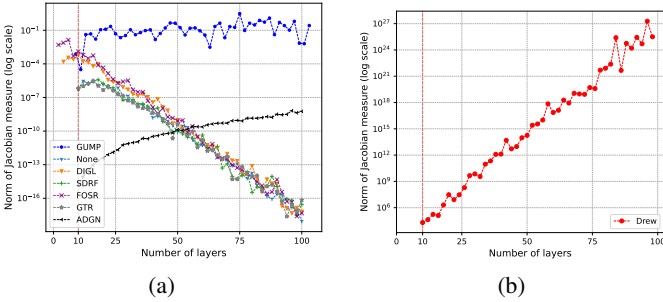

Figure 6: Jacobian measure versus layers on NCI1

**Performance on node classification tasks** We also apply GUMP to node classification tasks on Cora and Citeseer datasets. The results are shown in Table 8.

Table 8: Accuracy of node classification datasets: Cora and Citeseer

|  | Layers | 2 | 4 | 8 | 16 | 64 |
|---|---|---|---|---|---|---|
| Cora | GCN | 81.1 | 80.4 | 69.5 | 60.3 | 28.7 |
|  | GCNII | 82.2 | 82.6 | 84.2 | 84.6 | 85.5 |
|  | GCN-GUMP | 84.6 | 86.2 | 84.8 | 85.4 | 87.4 |
| Citeseer | GCN | 70.8 | 67.6 | 30.2 | 18.3 | 20.0 |
|  | GCNII | 68.2 | 68.9 | 70.6 | 72.9 | 73.4 |
|  | GCN-GUMP | 73.0 | 73.0 | 72.8 | 72.4 | 75.8 |

**Training time of GUMP**   The time of training GCN and GCN-GUMP 100 epochs on the TUDataset is shown in Table 9.

Table 9: Training seconds of GCN and GCN-GUMP on TUDataset for 100 epochs

|  | MUTAG | Proteins | Enzymes | NCI1 | NCI109 |
|---|---|---|---|---|---|
| GCN | 4.39 | 20.57 | 11.26 | 71.79 | 74.56 |
| GCN-GUMP | 23.26 | 228.44 | 972.64 | 615.17 | 637.48 |

**Results following Tönshoff et al. (2023)**   We compare GCN-GUMP with GCN on Peptides-func and Peptides-struct datasets following Tönshoff et al. (2023). The results are shown in Table 10.

Table 10: Follow Tönshoff et al. (2023) for comparison on Peptides-func and Peptides-struct.

|  | Peptides-func | Peptides-struct |
|---|---|---|
| GCN | 0.6860±0.0050 | 0.2460±0.0007 |
| GCN-GUMP | **0.6985±0.0032** | **0.2438±0.0014** |

**Average time of preprocessing**   The average time of preprocessing for line graph on various datasets is shown in Table 11.

Table 11: Average times (seconds) of preprocessing line graph for various dtasets

|  | MUTAG | Proteins | Enzymes | NCI1 | NCI109 | Peptides-func | Peptides-struct |
|---|---|---|---|---|---|---|---|
| Avg. Time | 0.001 | 0.005 | 0.009 | 0.002 | 0.002 | 0.014 | 0.013 |

## F   APPLY GUMP TO DIRECTED GRAPH

GUMP can also be applied to directed graphs in Algorithm 4. The transformation of directed graphs is also based on Lemma D.2.

---

**Algorithm 4** Graph transformation for directed graphs

---

**Require:** A directed graph $G = (V, E)$;
 1: Initialize a new digraph $G' = (V, E')$;
 2: **for** $(i, j) \in E$ **do**
 3:     Add $(i, j)$ and $(j, i)$ to $E'$;
 4: **end for**
 5: Remove duplicated edges in $E'$;
 6: Convert $G'$ to its line graph $\mathsf{L}(G')$;
 7: **Return:** A digraph $\mathsf{L}(G')$.

---

## G  KEY CODE SNIPPETS

**Code for obtaining $\mathbf{P}_1$ and $\mathbf{P}_2$ in Algorithm 2**    $\mathbf{P}_1$ and $\mathbf{P}_2$ are not explicitly derived in implementation and are pre-compute in the data preprocessing phase. The code snippet in Listing 1 shows how to transform the adjacency matrix to be block diagonal.

```python
def get_permutation_index(self, edge_index):
    num_nodes = edge_index.max() + 1
    permutation_index = []
    block_size = []
    node_flag = np.zeros(num_nodes)
    print(num_nodes)
    for i in tqdm(range(num_nodes)):
        if node_flag[i] == 0:
            edge_index_i = edge_index[1, edge_index[0] == i]
            edge_index_selected = edge_index[:, np.isin(edge_index[1, :],
     edge_index_i)]
            node_flag[edge_index_selected[0, :]] = 1
            permutation_index.append(edge_index_selected)
            block_size.append(edge_index_selected.shape[1])
    permutation_index = torch.cat(permutation_index, dim=1)
    return (permutation_index, block_size)
```

Listing 1: Code for pre-computing $\mathbf{P}_1$ and $\mathbf{P}_2$ in Algorithm 2

**Code for unitary projection in Algorithm 2**    The code snippet in Listing 2 shows how to calculate the unitary projection of the adjacency matrix.

```python
def proj(self, data):
    src_attn_x = data.x[data.edge_index_2[0]]
    dst_attn_x = data.x[data.edge_index_2[1]]
    x_src = self.gump_attn_src(src_attn_x)
    x_dst = self.gump_attn_dst(dst_attn_x)
    x_attn = (F.leaky_relu(x_src + x_dst, 0.2) * self.gump_attn_ele).sum(
    dim=-1)
    alpha = torch.tanh(x_attn)

    split_data = torch.tensor_split(alpha, data.blocksize[0, :-1].tolist
    ())
    A_sizes = np.array([A.shape[0] for A in split_data])
    sort_index = np.argsort(A_sizes)

    inv_sort_index = np.zeros_like(sort_index)
    inv_sort_index[sort_index] = np.arange(len(sort_index))

    sort_A_sizes = A_sizes[sort_index]
    segement_index = np.flatnonzero(np.diff(sort_A_sizes)) + 1
    segement_index = np.concatenate([[0], segement_index, [len(
    sort_A_sizes)]])

    sorted_split_data = [split_data[i] for i in sort_index]
    unitary_weight = []
    for i in range(len(segement_index) - 1):
        start, end = segement_index[i], segement_index[i+1]
        u_weight = torch.stack(sorted_split_data[start:end], dim=0)
        result = unitary_proj(u_weight, self.training)
        unitary_weight = unitary_weight + [ii for ii in result]
    unitary_alpha = [unitary_weight[i] for i in inv_sort_index]
    unitary_alpha = torch.cat(unitary_alpha)

    return unitary_alpha
```

Listing 2: Code for unitary projection in Algorithm 2

