# OpenReview forum: "GUMP: Alleviating Oversquashing with Unitary Message Passing"
_ICLR.cc/2025/Conference — Submitted to ICLR 2025_

### Official Review · Reviewer_Av26 · 2024-10-25

**Soundness:** 2
**Presentation:** 3
**Contribution:** 1
**Rating:** 5
**Confidence:** 3

**Summary:**

In this paper, the authors propose to address oversquashing in GNNs by imposing message-passing with a unitary matrix. Observing that most previous works modify the underlying connectivity of the graph, they propose a method that first consist in transforming the graph into its line graph, for which a theoretical result exists stating that there exists weighted adjacency matrices that are unitary. Then, they compute a new weighted adjacency matrix with attention mechanisms, then project it onto the set of unitary matrices by putting its singular values to 1. They give an efficient method for the last part if the adjacency is block-diagonal.

**Strengths:**

- an interesting idea
- good experimental results

**Weaknesses:**

The principles behind the method left me thoroughly confused. It seems immensely complicated, and just "obtaining unitarity" is not enough of a justification for all the steps involved in the method.

- Mainly, the authors remark that transforming a graph into its line graph proves that **there exists** unitary adjacency matrices with the same support than the original one. But, unless I am missing something, the projection of $\bar A$ onto the set of unitary matrices has no reason to respect this support condition. And vice-versa, if the support condition is not crucial, why go to the line graph at all, which is costly? One could have simply computed $A(A^\top A)^{-1}$ with the original adjacency matrix, resulting in also a unitary matrix.

- I don't get why we need the attention coefficients here. Could the simple adjacency matrix be used? (or maybe, one could imagine that computing weights allows to precisely find the unitary matrix that respect the support, but some kind of iterative projection algorithm must be envisioned here)

** Edit after rebuttal **
The authors response makes sense. I still have doubts about the method and some points would really deserve more clarity, but I raised my initial score.

- Concerning the efficient algorithm to put the singular values to 1, it would be good to describe how the line graph matrix is guaranteed to be block diagonal. Also, the algorithm requires to find a permutation to find the block diagonal structure, can this be done efficiently?

**Questions:**

See above, I have questions about the method itself, maybe I am missing something.

---

> ### Author Response · Authors · 2024-11-23
>
> Thank you for your valuable feedback. We address your questions and discuss your concerns point by point below. We are open to further discussions and willing to make any additional changes if needed.
>
> ### Weaknesses
>
> **W1.1:** mainly, the authors remark that transforming a graph into its line graph proves that there exists unitary adjacency matrices with the same support than the original one. But, unless I am missing something, the projection of $\bar{A}$ onto the set of unitary matrices has no reason to respect this support condition.
> > **R1.1:** The projection of $\bar{A}$ onto the set of unitary matrices respects the support condition because of Lemma 3.2 and Proposition 2.4. There is an illustrative example of our method in Figure 4 of Appendix A.
>
> **W1.2:** if the support condition is not crucial, why go to the line graph at all, which is costly? One could have simply computed $A(A^\top A)^{-0.5}$ with the original adjacency matrix, resulting in also a unitary matrix.
> > **R1.2:** The support condition contains the edge connectivity of the original graph and is crucial for graph learning [1]. Simply computing $A(A^\top A)^{-0.5}$ will lose the graph structure information.
> >
> > [1] Relational inductive biases, deep learning, and graph networks. 2018
>
> **W2.1:** I don't get why we need the attention coefficients here. Could the simple adjacency matrix be used?
> > **R2.1:** If using a simple adjacency matrix, the Jacobian measure will change exponentially, causing the oversquashing problem again.
>
> **W2.2:** one could imagine that computing weights allows to precisely find the unitary matrix that respect the support, but some kind of iterative projection algorithm must be envisioned here
> > **R2.2:** Unitary adjacency matrix depends on the input graph and the attention coefficients, learned from the graph, are data-dependent. Directly computing a unitary matrix with iterative projection is not robust because it is hard to design the initialization for the iterative projection. For example, if two graphs are isomorphic, the iterative projection may converge to different unitary matrices because of different initializations. The attention coefficients learned from the graph are permutation equivariant, which can guarantee the same unitary matrix for isomorphic graphs.
>
> **W3.1:** concerning the efficient algorithm to put the singular values to 1, it would be good to describe how the line graph matrix is guaranteed to be block diagonal.
> > **R3.1:** The line graph matrix can be permuted to be block diagonal because of Proposition 2.2 (See the proof in Appendix D). The proof is based on graph theory and is challenging to explain in simple terms. We have an illustrative example in Figure 4 of Appendix A to show the block diagonal structure of the line graph matrix.
>
> **W3.2:** the algorithm requires to find a permutation to find the block diagonal structure, can this be done efficiently?
> > **R3.2:** The preprocessing time is negligible and can be parallelized. We report the average preprocessing time (seconds) of one graph in the following table. We also presented the code for preprocessing in the `Listing 1` of Appendix G.
> > |             |     MUTAG    |     Proteins    |     Enzymes         |     NCI1      |     NCI109    | Peptides-func | Peptides-struct |
> > |-------------|--------------|-----------------|----------------|---------------|---------------|---------------|-----------------|
> > |     Avg. Time     |     0.001     |     0.005      |     0.009      |     0.002     |     0.002   | 0.014 | 0.013 |

---

### Official Review · Reviewer_aBWj · 2024-11-02

**Soundness:** 3
**Presentation:** 3
**Contribution:** 3
**Rating:** 6
**Confidence:** 4

**Summary:**

The work proposes to substitute the adjacency matrix of the graph with a related unitary adjacency matrix to alleviate issues such as over-squashing — in a permutation equivariant manner. The authors theoretically show the advantage of this method in terms of over-squashing sensitivity bounds and show experimentally its good performance.

**Strengths:**

I enjoyed the approach of the work. Many works that tackle over-squashing have a pretty similar flavour and usually propose to modify the connectivity of the graph. I think that this approach instead is quite novel and well-motivated. I enjoy the connection the work has to well-established ideas from the RNN literature and overall I found the idea to be quite sound.

The work also compares against a large number of baselines and overall has a very good range of experiments.

The presentation is also good with a lot of implementation details as well and a number of interesting discussions.

**Weaknesses:**

This is a small nitpick, but I believe that the “Jacobian measure” should be formally defined before the introduction of Theorem 2.1. It might be also better to leave the theorem more formally as I don’t think it’s too long to write out.

Please see questions.

**Questions:**

(Q1) I would like to have more clarity on the mathematical contributions of the work. For instance, is Proposition 2.2 a novel result or is this already known? More precisely, I am interested in understanding whether the propose idea of taking an adjacency matrix and constructing a related unitary matrix is a new idea in itself.

(Q2) Would it be possible to share the runtimes of GCN-GUMP against regular GCN? I read in the limitations that GUMP incurs a high computational cost, but I would be interested in understanding what this means in practice. I do not see the scalability as a weakness as I think that the idea is sufficiently interesting in itself, but I am curious to see the effiency of the implementation.

---

> ### Author Response · Authors · 2024-11-23
>
> Thank you for your valuable feedback on improving our work. We address your questions and discuss your concerns point by point below. We have made revisions to the paper based on your suggestions. We are open to further discussions and willing to make any additional changes if needed.
>
> ### Weaknesses
>
> **W1:** “Jacobian measure” should be formally defined before the introduction of Theorem 2.1
> > **R1:** We formally define “Jacobian measure” in line 153 page 3 and make Theorem 2.1 formally in revision. Thank you for the suggestions, we hope the revision will make the theorem more accessible to readers.
>
> ### Questions
>
> **Q1:** is Proposition 2.2 a novel result or is this already known?
> > **A1:** Proposition 2.2 is a novel result, which is derived based on results from [1].
> >
> > [1] Severini, Simone. "On the digraph of a unitary matrix." SIAM Journal on Matrix Analysis and Applications 25.1 (2003): 295-300.
>
> **Q2:** share the runtimes of GCN-GUMP against regular GCN
> > **A2:** In Table 9 of our submission, we reported the runtimes. The high computational cost of GUMP is attributed to the unitary projection, which is essential for GUMP to achieve an optimal Jacobian measure. In future work, we will explore using nearly unitary matrices to reduce this computational cost.

---

> > ### Comment · Reviewer_aBWj · 2024-11-23
> > **Thanks!**
> >
> > Thank you for your reply and clarifications + the additional table.
> >
> > I think that the work is interesting and I am happy to keep my score. I am following the discussion with reviewer Rfyx and interested in the conclusions from that discussion as well as I believe the reviewer raises some good points.

---

> > > ### Author Response · Authors · 2024-11-25
> > > **Acknowledgment**
> > >
> > > Dear Reviewer aBWj,
> > >
> > > We greatly appreciate your recognition of our work, which motivates us to continue innovative research in GNN. We also appreciate your attention to the discussion phase. Our discussion with Reviewer Rfyx is still ongoing, and his valuable feedback has made the "same support" property more accessible and improved the precision of paper writing.
> > >
> > > Sincerely,
> > >
> > > All authors

---

### Official Review · Reviewer_Rfyx · 2024-11-03

**Soundness:** 2
**Presentation:** 3
**Contribution:** 2
**Rating:** 3
**Confidence:** 3

**Summary:**

The paper explores the unitary message passing for alleviating oversquashing in GNNs. It first looks at the adjacency matrix of the line graph and then aims to find a unitary matrix with the same support as the line graph adjacency matrix. There are also some experiments to show the effectiveness of the proposed method.

**Strengths:**

Modification of a message passing matrix to be unitary is well motivated. The line graph approach seems novel and interesting.

**Weaknesses:**

1. There is some disconnection between Proposition 2.2 that the adjacency matrix of the line graph has the same support of some unitary matrix and the proposed method which finds the projection of a weighted adjacency matrix to the set of unitary matrices. It is unclear to me if the result in Proposition 2.3 has the same support as the adjacency matrix of the line graph.
2. The computational complexity of the proposed method is very high as it involves taking the square root of a matrix of the size $2e \times 2e$ where $e$ is the number of edges in the graph. Though the block matrix structure can be exploited, but there is no guarantee how many blocks can be found in the matrix. For example, it's likely that the proposed method cannot perform on all of the LRGB datasets.
3. The experiments are not very convincing. They only compared with the one-hop variant of some models that aims to solve the oversquashing problem. Note that the oversquashing problem is intrinsically multi-hop and I don't see the rationale weakening the baseline models to one-hop.
4. The preprocessing time that involves the computation of the block matrix is not reported.
5. It's unclear why there is a base layer GNN encoding in the proposed method. An ablation study on the necessity of the base layer GNN encoding would be helpful.
6. On the Peptide dataset, the GCN can easily achieve the accuracy of the proposed method by some proper data preprocessing or normalization. The authors should provide a comparison following [1].

[1] Tönshoff, Jan, et al. "Where did the gap go? Reassessing the long-range graph benchmark." arXiv preprint arXiv:2309.00367 (2023).

**Questions:**

See weakness

---

> ### Author Response · Authors · 2024-11-23
>
> Thank you for your valuable feedback. We answer your questions and discuss your concerns point by point below. We revised the paper according to your suggestions. We are open to further discussions and are willing to make additional changes if necessary.
>
> ### Weaknesses
>
> **W1:** disconnection between Proposition 2.2 and the proposed method. It is unclear if the result in Lemma 2.3 has the same support as the adjacency matrix of the line graph.
> > **R1:** In Proposition 2.2 and Lemma 2.3, the line graph $\mathsf{L}(G^\prime)$ indicates that they share the same support. To make the connection more explicit, we add more details at the beginning of Section 2.3 in the revision.
>
> **W2:** high computational complexity
> > **R2:** The high computational complexity of GUMP is attributed to the unitary projection. Although the unitary projection introduces significant computational complexity, it is essential for GUMP to obtain optimal Jacobian measure (as stated in Theorem 2.1) and exhibit good performance (as demonstrated in the ablation study in Figure 3c).
> >
> > In practice, GUMP is a good option for many tasks, particularly in biology and chemistry. Many graphs in these tasks are not very large, and the data collection and analysis processes often take significantly more time than model training. Considering the potential benefits GUMP offers in these tasks (such as drug discovery and chemical reaction prediction), the increase in time compared to baselines is deemed acceptable.
>
> **W3:** only one-hop message passing is compared.
> > **R3:** In Section 3.2, we explained the reasons for comparing single-hop variants. We supplement more reasons here:
> > - Studying one-hop MP is fundamentally important, as one-hop MP is the building block of GNNs and multi-hop MP.
> > - Comparing against multi-hop message passing is unfair because it is more powerful than one-hop MP. Multi-hop MP captures long-range dependencies from both multi-hop messages aggregation and layer stacking, while the one-hop MP learns the long-range dependencies only by stacking layers.
> > - By focusing on one-hop MP, we can eliminate interference from other factors and better understand the impact of oversquashing.
>
> **W4:** preprocessing time of the block matrix.
> > **R4:** We report the average preprocessing time (seconds) of one graph in the following table. The preprocessing time is negligible and preprocessing can be parallelized. We also present the code for preprocessing in the `Listing 1` of Appendix G.
> > |             |     MUTAG    |     Proteins    |     Enzymes         |     NCI1      |     NCI109    | Peptides-func | Peptides-struct |
> > |-------------|--------------|-----------------|----------------|---------------|---------------|---------------|-----------------|
> > |     Avg. Time     |     0.001     |     0.005      |     0.009      |     0.002     |     0.002   | 0.014 | 0.013 |
>
> **W5:** ablation study on the necessity of the base layer GNN encoding.
> > **R5:** The base GNN is used for processing the input graph and obtaining the initial node embeddings. We conducted an ablation study on the necessity of the base GNN in Figure 3(c) (w/o base GNN) of submission. The results show that the performance of GUMP varies on different datasets, i.e., the performance on NCI1 decreases, while the performance on Proteins does not. This phenomenon is expected because the quality of the unitary adjacency matrix depends on the representations of nodes in the line graph.
>
> **W6:** provide a comparison following [1] for the Peptide dataset.
> > **R6:** We apply the data preprocessing or normalization in [1] to the Peptide dataset and compare the performance of GUMP and GCN. The results show that GCN-GUMP outperforms GCN on the Peptide dataset. We present the results in the following table.
> > |  | Peptides-func | Peptides-struct |
> > |----------|----------|----------|
> > | GCN | 0.6860±0.0050 |  0.2460±0.0007  |
> > | GUMP | 0.6985±0.0032 | 0.2438±0.0014 |

---

> > ### Comment · Reviewer_Rfyx · 2024-11-23
> >
> > Thank you for the effort. Before further discussion, can you prove why the unitary/orthogonal projection of a weighted adjacent matrix of the line graph has the same support?

---

> > > ### Author Response · Authors · 2024-11-23
> > >
> > > Thank you for the valuable feedback. We add Corollary 2.5 (proved in Appendix D.4) to make the conclusion and proof more accessible in the revision. The conclusion is a result based on Proposition 2.4 (or Proposition 2.6). The basic ideas of the proof are as follows:
> > >
> > > 1. The line graph $\mathsf{L}(G^\prime)$ has a special property that its support can be permuted by rows and columns to be a block diagonal matrix $\mathbf{D}=\textsf{diag}\\{\mathbf{D_1}, \cdots, \mathbf{D_b}\\}$ with each block $\mathbf{D}_i$ a dense support, i.e., $\mathbf{D_i}=\mathbf{1}$. We assume the permutation matrices to be $\mathbf{P}_1$ and $\mathbf{P}_2$ for rows and columns, respectively. (See Lemma D.4)
> > > 2. For each dense sub-matrix $\mathbf{D}_i$ from the block diagonal matrix, there are unitary matrices with support $\mathbf{D}_i$. Therefore, there are unitary matrices (i.e., $\mathsf{U}[\mathbf{P}_1 \bar{\mathbf{A}}\mathbf{P}_2^\top]$ in our paper) with support $\mathbf{D}$.
> > > 3. Since row and column permutations are invertible, we can permute the support of the unitary matrix back to the original support of the line graph $\mathsf{L}(G^\prime)$, i.e., $\mathbf{P}_1^\top \mathsf{U}[\mathbf{P}_1 \bar{\mathbf{A}}\mathbf{P}_2^\top] \mathbf{P}_2$ has the same support as $\mathbf{D}$. Because the permutation matrices $\mathbf{P}_1$ and $\mathbf{P}_2$ are unitary, the matrix $\mathbf{P}_1^\top \mathsf{U}[\mathbf{P}_1 \bar{\mathbf{A}}\mathbf{P}_2^\top] \mathbf{P}_2$ is also a unitary matrix.
> > > 4. In Proposition 2.4, we show that $\mathsf{U}[\bar{\mathbf{A}}]=\mathbf{P}_1^\top \mathsf{U}[\mathbf{P}_1 \bar{\mathbf{A}}\mathbf{P}_2^\top] \mathbf{P}_2$. Thus, the unitary/orthogonal projection of a weighted adjacent matrix of the line graph has the same support.
> > >
> > > We hope our response clarifies the proof of the same support. We are grateful for your feedback and are open to further discussions.

---

> > > > ### Comment · Reviewer_Rfyx · 2024-11-23
> > > >
> > > > Thank you for your quick reply. It seems that you are implicitly assuming that the orthogonal projection of a full support matrix must have full support, which is not clear to me. For example, we can perturb the identity matrix in a direction that is orthogonal to the tangent space. For example, consider making the off-diagonal matrix a small positive number while maintaining the diagonal entries as one. Then, if such a perturbed matrix occurs as a block, then its orthogonal projection is the identity that is not fully supported.
> > > >
> > > > Also, since the attention matrix only has real numbers, the unitary matrix or projection considered in this paper is essentially orthogonal. If so, why use utility when all you need is orthogonal?

---

> > > > > ### Author Response · Authors · 2024-11-23
> > > > >
> > > > > Thank you for your reply. We are glad to have further discussion with you. We will address your concerns point by point below.
> > > > >
> > > > > **Q1:** implicit assumption that the orthogonal projection of a full support matrix must have full support.
> > > > > > **A1:** We appreciate your insightful and interesting question. We can mathematically describe your problem as
> > > > > >> *Problem:* Assuming $\mathbf{A}$ is a dense matrix, when $\mathbf{A}\left( \mathbf{A}^\dagger \mathbf{A} \right)^{-\frac{1}{2}}$ is not dense?
> > > > > >
> > > > > > Actually, there is no general answer to this problem and it is an open problem in algebra. However, we observe that when $\mathbf{A}$ is a dense matrix, the projection $\mathbf{A}\left( \mathbf{A}^\dagger \mathbf{A} \right)^{-\frac{1}{2}}$ is likely to be dense unless $\mathbf{A}$ possesses certain special structures. Our understanding of this problem stems from the following two observations:
> > > > > > - **O1:** The multiplication of two dense matrices is likely to be dense.
> > > > > > - **O1:** The inverse square root of a dense matrix is likely to be dense.
> > > > > >
> > > > > > Based on **O1** and **O2**, it is probable that $\mathbf{A}^\dagger \mathbf{A}$ and its inverse square root $(\mathbf{A}^\dagger \mathbf{A})^{-\frac{1}{2}}$ are dense in the projection. Consequently, due to **O1**, the projection $\mathbf{A}\left( \mathbf{A}^\dagger \mathbf{A} \right)^{-\frac{1}{2}}$ is also likely to be dense, which is consistent with our empirical observations.
> > > > > >
> > > > > > Regarding your example, the orthogonal projection of an identity matrix with small noisy off-diagonal entries is still fully supported (the projected matrix is close to an identity matrix). Here is a simple code that can validate this:
> > > > > > ```python
> > > > > > import numpy as np
> > > > > > from scipy.linalg import fractional_matrix_power
> > > > > >
> > > > > > I = np.eye(4)
> > > > > > N = np.random.rand(4, 4)*1e-4
> > > > > > np.fill_diagonal(N, 0)
> > > > > > A = I + N
> > > > > > B = A @ fractional_matrix_power(A.T @ A, -0.5)
> > > > > > print("Matrix B:\n", B)
> > > > > > ```
> > > > >
> > > > > **Q2:** why use unitary when all you need is orthogonal?
> > > > > > **A2:** We have the following reasons for using "unitary":
> > > > > > - Since orthogonal matrices are unitary matrices, using "unitary" does not affect the correctness of our paper.
> > > > > > - Because our paper is motivated by unitary RNNs, e.g., [1,2], we use "unitary" to emphasize the connection between our work and unitary RNNs.
> > > > > > - Technically, GUMP can work with unitary matrix with complex entries. However, the significance of complex-valued GNN is still an open question, so we do not explore it in this paper.
> > > > > > - Finally, the term "GUMP" is catchier and more memorable than "GOMP" (Graph Orthogonal Message Passing) because of a famous movie character.
> > > > > >
> > > > > > [1] Arjovsky, Martin, Amar Shah, and Yoshua Bengio. Unitary evolution recurrent neural networks. ICML 2016.
> > > > > >
> > > > > > [2] Kiani, Bobak, Randall Balestriero, Yann LeCun, and Seth Lloyd. projUNN: efficient method for training deep networks with unitary matrices. NeurIPS 2022

---

> > > > > > ### Comment · Reviewer_Rfyx · 2024-11-23
> > > > > >
> > > > > > "likely" is not enough to support the proof of Corollary 2.5, and the main claim of the paper that using projection always leads to a unitary matrix with the same support is unfounded and one can find counter-examples.
> > > > > >
> > > > > > My suggestion was to perturb the identity matrix in a **normal direction** to the tangent space of the unitary matrix so that it can be guaranteed that the identity matrix is the projection of the perturbed matrix. Since the tangent space of the unitary matrix at the identity is formed by matrices with $A^T + A= 0$. Then, for example, we can choose any symmetric matrix $N$ for the perturbation matrix. In this case, $N$ is symmetric. As $A$ is $I + N$, it will also be symmetric. Then the inverse square root of $A^TA$ is just $A^{-1}$, and the projection is an identity matrix that is sparse.
> > > > > >
> > > > > > Your idea of using unitary sounds interesting, but I believe precision in scientific writing should be prioritized over other considerations.

---

> ### Author Response · Authors · 2024-11-24
>
> Thank you for the valuable feedback. We revised the paper according to your suggestions. We move Corollary 2.5 to Section 2.3.2 and add the assumption of fully supported unitary projected matrix:
> > **Corollary** With the strong permutation equivariance in Proposition 2.4, assuming each $\mathsf{U}[\mathbf{D}_i]$ is fully supported, the matrix returned by Algorithm 2 has the same support as the line graph $\mathsf{L}(G^\prime)$.
>
> The rationality of this assumption is in the following aspects:
> - The block diagonal matrix from $\mathbf{P}_1\bar{\mathbf{A}}\mathbf{P}_2^\top$, composed of attention weights, does not have the special structures to be projected to be sparse.
> - In our experiments, we find that the support of $\mathsf{U}[\bar{\mathbf{A}}]$ and $\mathsf{L}(G^\prime)$ are consistent.
>
> We hope our response can address your concern. We greatly appreciate your feedback and are glad for further discussions.

---

### Official Review · Reviewer_2i6Y · 2024-11-10

**Soundness:** 4
**Presentation:** 3
**Contribution:** 3
**Rating:** 6
**Confidence:** 3

**Summary:**

The paper addresses the oversquashing problem in GNNs by introducing Graph Unitary Message Passing , which uses a unitary adjacency matrix for message passing. Unlike previous methods that rely on rewiring and disrupt original graph connectivity, GUMP maintains the graph’s structure while alleviating oversquashing. The approach involves a transformation to equip graphs with unitary adjacency matrices and a unitary projection algorithm, ensuring permutation-equivariance. Extensive experiments show that incorporating GUMP into various GNN architectures improves performance across multiple graph learning tasks, demonstrating its effectiveness in tackling oversquashing.

**Strengths:**

Novelty:
The idea is innovative, building upon unitary RNNs to create Graph Unitary Message Passing (GUMP) as a solution to the oversquashing problem in GNNs. The approach is grounded in a solid theoretical foundation, demonstrating a reasonable and well-motivated extension to graph learning.

Empirical Results:
The paper demonstrates strong empirical results, showing significant improvements across multiple datasets compared to prior methods. This highlights the practical effectiveness of GUMP in addressing oversquashing and enhancing graph learning tasks.

Clarity and Analysis:
The Jacobian measure is clearly presented, and the results show that it does not decrease, providing robust evidence of the method's impact. The ablation study is well-executed, delivering clear and actionable insights into the contributions of various components of the method.

Novel Construction Approach:
The paper introduces a novel and elegant method to construct equivalent graphs and compute unitary adjacency matrices. This approach maintains the original graph connectivity while ensuring permutation-equivariance, representing a creative and impactful contribution to the field.

**Weaknesses:**

The paper would benefit from a clearer introduction to unitary matrices, their role in RNNs, for the readers without enough background.

**Questions:**

Theorem 2.1 is highly important. Could you also briefly clarify the differences between the informal theorem and the formal version presented in the main content?

---

> ### Author Response · Authors · 2024-11-23
>
> Thank you for taking the time to review our work and provide constructive feedback. We answer your questions and discuss your concerns point by point below. We revised the paper according to your suggestions and believe that the revision will make the paper more accessible to readers. We are open to further discussions and are willing to make additional changes if necessary.
>
> ### Weaknesses
> **W1:** lack a clearer introduction to unitary matrices, and their role in RNNs.
> > **R1:** We have added more details about unitary matrices and their role in RNNs in Appendix B.2 of the revision.
>
> ### Questions
> **Q1:** Theorem 2.1 is highly important. Could you also briefly clarify the differences between the informal theorem and the formal version presented in the main content?
>
> > **A1:** The informal theorem and the formal version are essentially the same. To make the theoretical contribution more accessible, we remove the informal theorem and present the formal version in the main content of the revision.

---

### Author Response · Authors · 2024-11-23
**Summary of Revision**

We thank all reviewers for their valuable feedback. In summary, we have made the following revisions to the paper:

- We remove the informal theorem and present the formal version (Theorem 2.1) in the main content. (Concerns of Reviewers 2i6Y and aBWj)
- We add more details about unitary matrices and their role in RNNs in Appendix B.2. (Concern of Reviewer 2i6Y)
- We include additional results on the new experimental setting (Reviewer Rfyx) and the preprocessing time of the block matrix (Reviewers Rfyx and Av26) in Appendix E.

We appreciate the reviewers' comments and suggestions and are open to further discussions.

---

### Meta-Review · Area_Chair_91xC · 2024-12-18

**Metareview:**

This paper proposes to address GNN oversquashing by employing a unitary matrix for message passing.
The reviewers claimed that this is an interesting idea. However, reviewer Rfyx found a technical mistake that required changing the assumptions of the main technical result. The issue was not fully resolved during the discussion period, so I suggest the authors revise the paper carefully and resubmit it elsewhere.

**Additional Comments On Reviewer Discussion:**

Please note the back-and-forth discussion with reviewer Rfyx. The reviewer found a technical mistake that the authors acknowledged and tried to resolve, but the reviewer wasn't in the end satisfied with the provided answer.

---

### Decision · Program_Chairs · 2025-01-22

Reject